# Understanding drivers of phylogenetic clustering and terminal branch lengths distribution in epidemics of *Mycobacterium tuberculosis*

**Fabrizio Menardo\***

Department of Plant and Microbial Biology, University of Zurich, Zurich, Switzerland

**Abstract** Detecting factors associated with transmission is important to understand disease epidemics, and to design effective public health measures. Clustering and terminal branch lengths (TBL) analyses are commonly applied to genomic data sets of *Mycobacterium tuberculosis* (MTB) to identify sub-populations with increased transmission. Here, I used a simulation-based approach to investigate what epidemiological processes influence the results of clustering and TBL analyses, and whether differences in transmission can be detected with these methods. I simulated MTB epidemics with different dynamics (latency, infectious period, transmission rate, basic reproductive number R0, sampling proportion, sampling period, and molecular clock), and found that all considered factors, except for the length of the infectious period, affect the results of clustering and TBL distributions. I show that standard interpretations of this type of analyses ignore two main caveats: (1) clustering results and TBL depend on many factors that have nothing to do with transmission, (2) clustering results and TBL do not tell anything about whether the epidemic is stable, growing, or shrinking, unless all the additional parameters that influence these metrics are known, or assumed identical between sub-populations. An important consequence is that the optimal SNP threshold for clustering depends on the epidemiological conditions, and that sub-populations with different epidemiological characteristics should not be analyzed with the same threshold. Finally, these results suggest that different clustering rates and TBL distributions, that are found consistently between different MTB lineages, are probably due to intrinsic bacterial factors, and do not indicate necessarily differences in transmission or evolutionary success.

**\*For correspondence:**
fabrizio.menardo@uzh.ch

**Competing interest:** The author declares that no competing interests exist.

## Editor's evaluation

Grouping pathogen genomes into clusters is a key tool in genomic epidemiology. In this paper, the author takes a simulation-based approach to investigate the epidemiological processes that influence clustering in tuberculosis genomic epidemiology. The simulations explore whether differences in transmission can be detected with clustering-based analysis. This work finds that clustering can be impacted by sampling strategy as well as by changes in transmission and population dynamics, and draws out some interpretations of these results for users of clustering in this field.

## Introduction

In the last decade well beyond half a million bacterial genomes have been sequenced worldwide, about 7% of these from *Mycobacterium tuberculosis* (MTB) strains (*Blackwell et al., 2021*). One of the reasons behind these extensive sequencing efforts is the use of whole genome sequencing in molecular epidemiology. Molecular epidemiology studies of MTB (and of other pathogens) use microbial

genome sequences sampled from different patients to investigate epidemiological dynamics such as transmission, relapses, and the acquisition and spread of antibiotic resistance (*Hatherell et al., 2016*; *Guthrie and Gardy, 2017*; *Nikolayevskyy et al., 2019*). One of the most popular approaches to analyze this data is to cluster strains in groups based on their genetic distance. The identification of clustered MTB strains is commonly interpreted as evidence for recent local transmission, while patients infected with non-clustered strains (singletons) are thought to be novel introductions (i.e. patients that got infected somewhere else). Similarly, when studying the epidemiology of antibiotic resistance, clustered strains are considered as cases of transmission of resistance, while singletons are thought to be more likely to represent instances of resistance acquisition (*Hatherell et al., 2016*).

In MTB studies, clusters are often defined based on a single nucleotide polymorphism (SNP) threshold: strains with fewer SNPs than a given threshold are grouped together in the same cluster. However, this has been criticized, because it is not clear which value should be used (*Stimson et al., 2019*). A review of more than 30 publications concluded that a threshold of six SNPs could be used to identify cases of direct transmission (*Nikolayevskyy et al., 2019*), but other studies proposed different thresholds for different settings (see Table 1 in *Hatherell et al., 2016* for some examples). Alternative approaches have been proposed to overcome the limits of SNP thresholds. For example, the method proposed by *Stimson et al., 2019* groups strains based on a probabilistic model that takes into account the clock rate and the time of sampling. Other studies sidestepped the choice of one specific value by performing clustering multiple times with different thresholds, and comparing the results (*Holt et al., 2018*; *Meehan et al., 2018*; *Yang et al., 2018*; *López et al., 2020*; *Shuaib et al., 2020*; *Cox et al., 2021*, *Liu et al., 2021*; *Walter et al., 2022*; *Yang et al., 2022*). Despite their limitations, clustering methods are considered useful to study transmission dynamics in MTB. Many studies tested the association of clustered strains with host and bacterial sub-populations, such as different age groups, HIV-positive patients, bacterial lineages and others (*Guerra-Assunção et al., 2015*; *Asare et al., 2020*; *Sobkowiak et al., 2020*; *Cox et al., 2021*, *Gygli et al., 2021*; *Merker et al., 2021*; *Yang et al., 2022*). In these analyses, a positive association is interpreted as evidence for increased transmission of a certain sub-population. Further studies used clustering rates (the percentage of clustered strains), to characterize the extent of transmission in bacterial sub-populations (*Holt et al., 2018*; *Shuaib et al., 2020*). For example, *Holt et al., 2018* found that in Vietnam, MTB Lineage 2 (L2) had higher clustering rates compared to MTB Lineage 4 (L4) and MTB Lineage 1 (L1), and interpreted these results as evidence for more frequent transmission of L2 strains, compared to L4 and L1 strains. Finally, in recent studies the distribution of terminal branch lengths (TBL) was used as a proxy for transmissibility in different MTB lineages (*Holt et al., 2018*; *Freschi et al., 2021*; *Walter et al., 2022*), partially complementing classical clustering analyses. All these approaches are based on the assumption that increased transmission results in increased clustering rates and shorter terminal branches. However, it is known that other factors could influence the results of clustering, for example it was posited that higher rates of molecular evolution, and low sampling rates should lead to lower clustering rates (*Stimson et al., 2019*, *Menardo et al., 2019*).

There is a consensus that epidemiological dynamics have an influence on the shape of MTB phylogenies, and therefore on clustering and TBL, though how they do so was never explored with quantitative studies. Here, I used simulations to explore what factors influence clustering results and TBL in MTB epidemics. I simulated the molecular evolution of MTB strains under different epidemiological conditions. I then inferred phylogenetic trees and computed clustering rates and TBL from the simulated data. With this approach I investigated the molecular clock rate, sampling proportion, sampling period, transmission rate, basic reproductive number (R0), length of the latency period, and length of the infectious period. I found that all these factors affect the results of clustering and TBL, except for the length of the infectious period. These results are in contradiction with the standard interpretation of MTB epidemiological studies, namely that sub-populations associated with clustering and shorter terminal branches are necessarily transmitting more.

## Results

To investigate the expected patterns of genetic diversity under different epidemiological conditions, I assembled a pipeline to simulate the evolution of MTB genomes in different epidemiological settings (*Figure 1*). The details are reported in the Materials and methods section. Briefly, the pipeline simulates a transmission tree using a birth-death model in which transmission events occur at rate $\lambda$ and

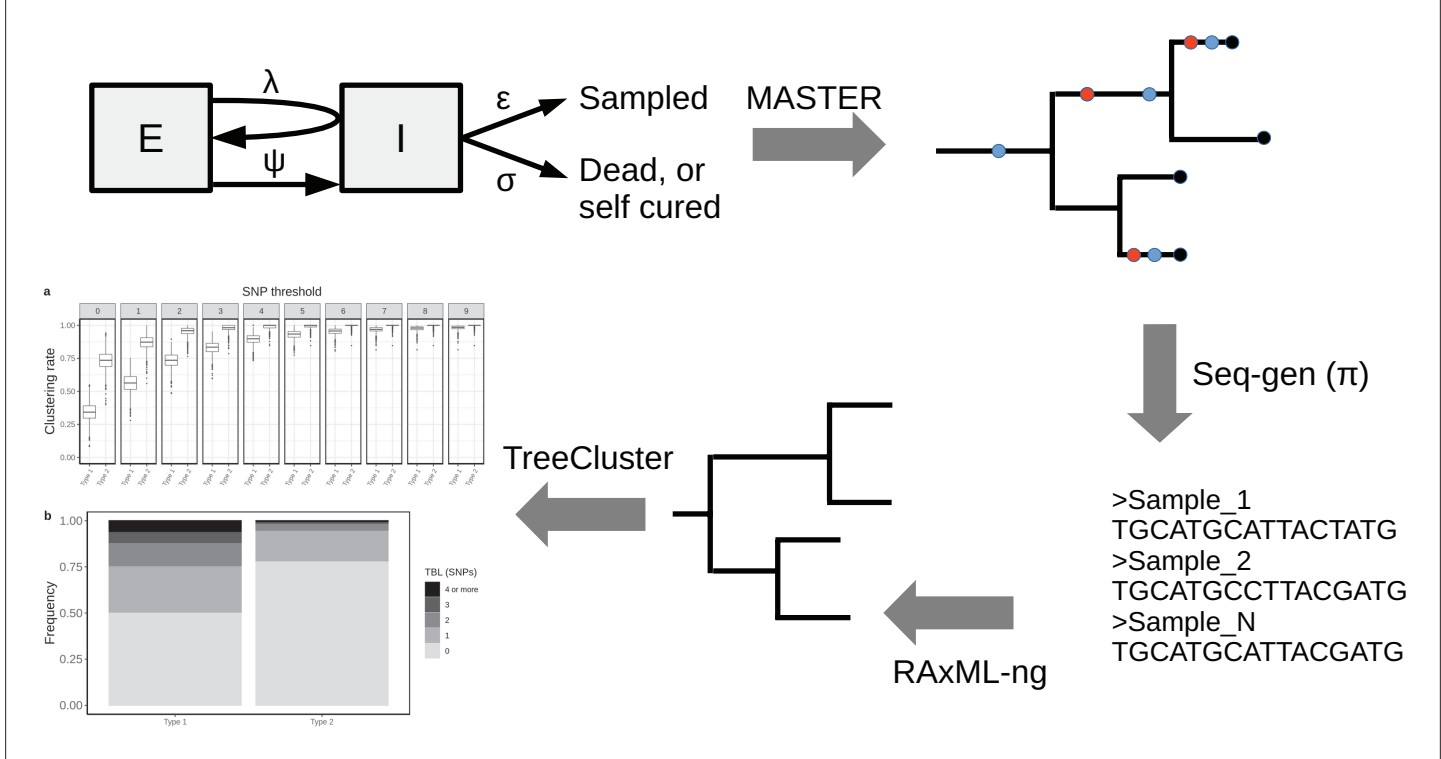

**Figure 1.** Simplified workflow, not all steps are depicted (see the Materials and methods section for details). A transmission tree is simulated by MASTER, given the represented epidemiological model and a set of parameters ($\lambda$, $\psi$, $\sigma$, and $\varepsilon$). Seq-gen is used to simulate the evolution of MTB genome sequences along the tree, given a clock rate ($\pi$). RAxML-ng is used to estimate the phylogenetic tree from the sequence data, and TreeCluster to perform clustering.

originate in the infectious compartment I, leading to a new exposed individual in compartment E. Exposed individuals become infectious at rate $\psi$, by moving from compartment E to compartment I. The time spent in the compartment E represents the latency period. Individuals are removed from compartment I by death or self-cure, occurring at rate $\sigma$, or sampling, occurring at rate $\varepsilon$. The sampling rate $\varepsilon$ corresponds to the rate at which infectious individuals are diagnosed and treated. With the onset of treatment, it is assumed that patients stop being infectious immediately. The time spent in compartment I represents the infectious period.

A similar epidemiological model was used previously in a phylodynamic analysis of two MTB outbreaks (**Kühnert et al., 2016**; **Kühnert et al., 2018**). In a second step, the pipeline simulates the evolution of genome sequences along the tree given a clock rate $\pi$, and it computes the clustering rates under different SNP thresholds, and the terminal branch lengths. This pipeline allows to test how clustering rates and TBL distributions change under different sampling proportions, sampling periods, molecular clock rates, transmission rates, basic reproductive numbers (R0 = $\lambda/(\sigma+\varepsilon)$), lengths of the latency period (determined by $\psi$), and lengths of the infectious period (determined by $\sigma+\varepsilon$). Moreover, with this approach it is possible to investigate how the different factors impact the choice of a sensible SNP threshold. To do this, I defined the '95% sensitivity SNP threshold' as the minimum threshold for which at least 95% of strains are clustered in at least 95% of the simulations. A lower SNP threshold would lead to lower sensitivity (i.e. simulated samples, which are the result of recent transmission, would not be clustered), while a larger threshold would lead to low specificity, although this cannot be quantified with this analysis (see Materials and methods for additional information).

## Clock rate, sampling proportion, and sampling period

First, I tested whether the clock rate, sampling proportion, and sampling period affect clustering rates and TBL. The details of these analyses are available as Appendix. I considered three different clock rates ($4 \times 10^{-8}$, $8 \times 10^{-8}$, and $1.2 \times 10^{-7}$ nucleotide changes per site per year), four sampling

proportions (25%, 50%, 75%, and 100% of cases sampled), and three sampling periods (5, 10, and 20 years). For each scenario, I performed 1000 simulations and compared clustering rates and TBL. I found that: (1) higher clock rates led to lower clustering rates and longer TBL (Appendix 1), (2) lower sampling proportions resulted in lower clustering rates and longer TBL (Appendix 2), and (3) shorter sampling periods resulted in lower clustering rates and longer terminal branches (Appendix 3). Finally, I also tested whether different sample sizes could have an influence on the results of these analyses, and found that TBL and median clustering rates did not change when using a lower threshold on the minimum number of tips in the simulated tree (Appendix 4).

## Latency

Next, I tested whether differences in the duration of the latent period could result in different clustering rates and TBL. Here, latency is defined as the period in which an individual is infected but not yet infectious. Typically, the shift to infectiousness in TB patients is considered to occur with the onset of symptoms. However, in recent years, the importance of sub-clinical TB has been reconsidered. It is possible that a considerable part of TB transmission occurs from asymptomatic patients, although this has not been yet quantified (*Kendall et al., 2021*). Given the uncertainty about the length of the latent period I tested three different rates of progression to infectiousness $\psi$=0.5, 1, 2, corresponding to a median latent period of ~16.6, 8.3, and 4.2 months, respectively. These values represent the range of duration of asymptomatic infection estimated in different countries (*Ku et al., 2021*). All other parameters were constant in all simulations ($\pi$=8 × 10$^{-8}$; $\sigma = \varepsilon$=0.5; $\lambda$ =1, sampling period = 10 years). I found that longer latency resulted in lower clustering rates and longer TBL (*Figure 2*, *Table 1*). Correspondingly, the 95% sensitivity threshold was 10, 6, and 5 SNPs, and the mean of the TBL distribution was 0.87, 0.56, and 0.41, respectively for long, mid, and short latency.

## Transmission, infectious period, and R0

I tested how different transmission and sampling rates impact the results of clustering and the TBL distributions. In these analyses, I set the death rate $\sigma$ to zero, so that all individuals are sampled, and the length of the infectious period is therefore determined by the sampling rate $\varepsilon$. I found that clustering results and TBL depend on the transmission rate, with higher values leading to higher clustering rates and shorter TBL. Conversely, the sampling rate had no major effect (Appendix 5). I tested whether the threshold on the minimum tree size could bias these analyses (the difference from Appendix 4 is that R0 can be different from one). I found that different thresholds on the minimum number of tips sampled in the simulated tree can influence clustering and TBL, but for the settings used in this study this effect was negligible, and the results were robust to different thresholds (Appendix 5).

Sampling and transmission rates are particularly interesting parameters because together they determine R0, and whether the epidemic will grow or shrink. R0 is the ratio between the rate at which infectious individuals are created (numerator) and the rate at which infectious individuals are removed (denominator). If the numerator is larger than the denominator the epidemic will grow (R0 >1). Conversely, if the denominator is larger than the numerator the epidemic will shrink (R0 <1). Specifically, with the epidemiological model used in this study the numerator is the transmission rate ($\lambda$), while the denominator is determined by the sum of the sampling and death rates ($\sigma$+$\varepsilon$), so that R0 = $\lambda$ / ($\sigma$ +$\varepsilon$). Because only the numerator affects the results of clustering rates and TBL, changes in these two metrics will correlate with R0 only when the denominator ($\sigma$ +$\varepsilon$) is fixed. To show this I tested three different sampling rates: $\varepsilon$=0.5, 1, and 2, corresponding to a median infectious period of ~16.6, 8.3, and 4.2 months, respectively. These values cover well the possible length of the symptomatic period estimated in different countries (*Ku et al., 2021*). For each value of $\varepsilon$, I considered three different transmission rates leading to three scenarios: a shrinking epidemic with R0=0.9, a stable epidemic with R0=1, and a growing epidemic with R0=1.1. All other parameters were constant in all simulations (*Table 2*). When considering scenarios with different sampling rates, and therefore infectious periods, I found no correlation between R0 and clustering rates, nor between R0 and TBL distributions. However, when the duration of the infection period, and all other parameters (except the transmission rate) were fixed, larger values of R0 correlated with higher clustering rates and shorter TBL (*Table 2*, *Figure 3*).

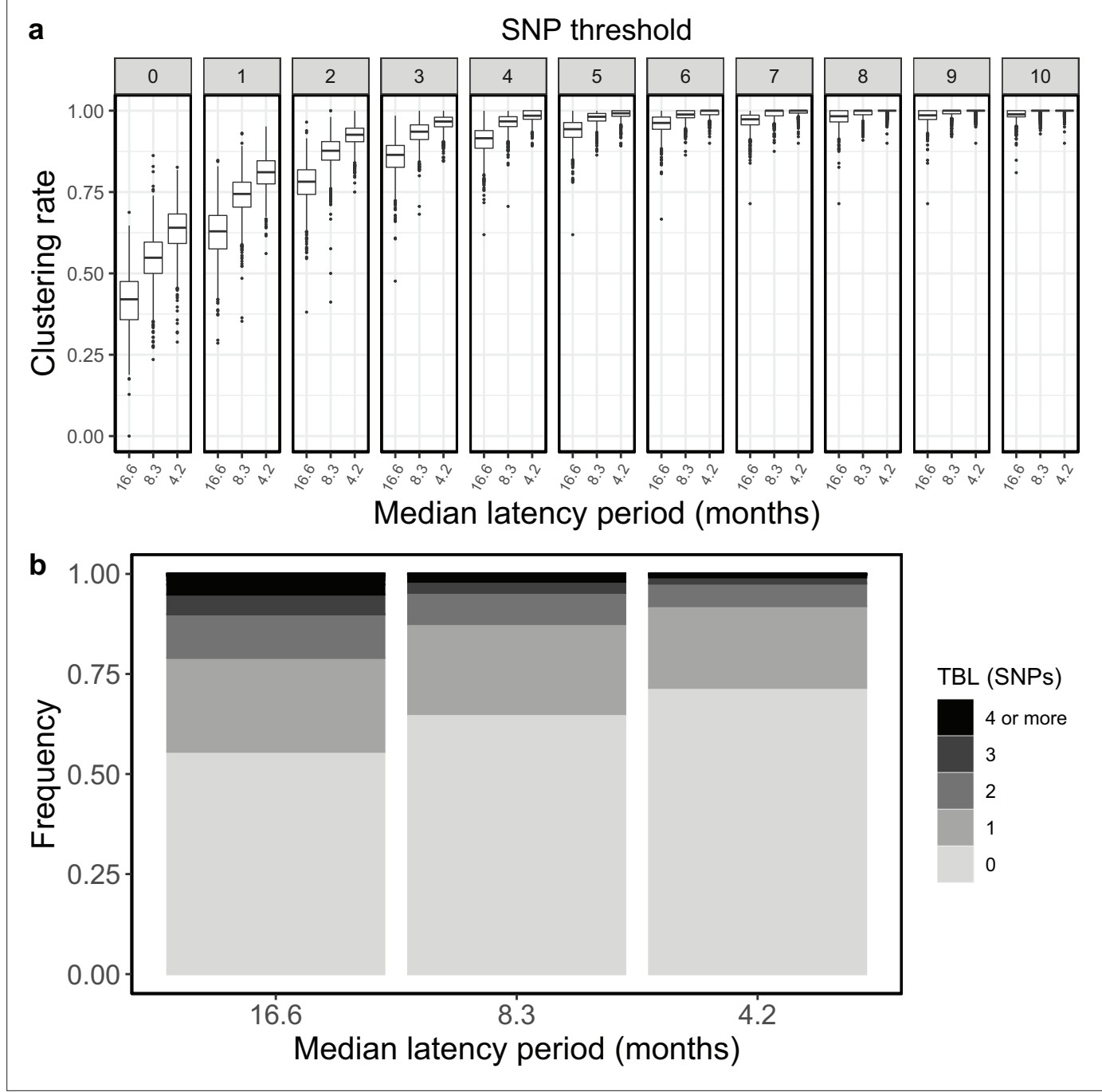

**Figure 2.** Clustering rates and TBL distributions for different rates of progression to infectiousness, and therefore latency period. (**a**) Clustering rates with different SNP thresholds. Only SNP thresholds up to the highest 95% sensitivity threshold are plotted (i.e. for thresholds higher than 10 SNPs more than 95% of samples are clustered in more than 95% of simulations for all settings). (**b**) Overall TBL distributions computed by merging all simulations.

## An example

The results reported above have important implications on the interpretation of molecular epidemiology studies of MTB. To illustrate this in practice let's consider an example in which two different strains of MTB are causing an epidemic in the same area. The two strains have different epidemiological and biological characteristics. We can think about it like two lineages of MTB, but it could also be two different clones belonging to the same lineage. MTB type 1 is expanding, with a R0 of 1.1, and it is characterized by a slow disease progression, with a median latency period of about one year. Type 1

**Table 1.** Parameters and results for the different simulated scenarios in the analysis of latency. $\lambda$: transmission rate, $\varepsilon$: sampling rate, R0 = $\lambda/(\varepsilon+\sigma)$, $\sigma$: death rate, $\psi$: rate of progression to infectiousness, $\pi$: molecular clock rate in expected nucleotide changes per site per year, 95% SNP threshold: the minimum SNP threshold for which at least 95% of samples are clustered in at least 95% of simulations, 100% SNP threshold: the minimum SNP threshold for which 100% of samples are clustered in at least 95% of simulations, 95% CI TBL: the confidence interval for the overall TBL distribution, Mean TBL: average of the overall TBL distribution.

| Scenario | $\lambda$ | $\varepsilon$ | R0 | $\sigma$ | $\psi$ | $\pi$ | 95% SNP threshold | 100% SNP threshold | 95% CI TBL | MeanTBL |
|---|---|---|---|---|---|---|---|---|---|---|
| Short latency | 1 | 0.5 | 1 | 0.5 | 2 | $8 \times 10^{-8}$ | 5 | 15 | 0–2 | 0.41 |
| Mid latency | 1 | 0.5 | 1 | 0.5 | 1 | $8 \times 10^{-8}$ | 6 | 17 | 0–3 | 0.56 |
| Long latency | 1 | 0.5 | 1 | 0.5 | 0.5 | $8 \times 10^{-8}$ | 10 | 20 | 0–5 | 0.87 |

populations are expected to increase by ~150% every 10 years. Conversely type 2 is shrinking, with a R0 of 0.9, and it is characterized by a shorter median latency period, approximately 5 months. Type 2 populations are expected to shrink by ~50% every 10 years. In addition, type 1 and 2 have moderate differences in their rate of molecular evolution, with a clock rate of respectively $1 \times 10^{-7}$ and $7 \times 10^{-8}$ nucleotide changes per site per year. In all other aspects the two types are identical. Obviously, under this scenario, type 1 is much more concerning for public health compared to type 2, at least in the long term. I repeated the same analysis presented above for the two types (*Table 3*).

Type 2 showed shorter terminal branches: the mean of the TBL distribution was 0.84 and 0.4 SNPs for type 1 and type 2, respectively. Moreover, type 2 had higher clustering rates (*Figure 4*), and the 95% sensitivity threshold was 8 and 5 SNPs for type 1 and 2, respectively. A typical interpretation of this data would be that type 2 is transmitting more than type 1, because of the shorter terminal branches, and the higher clustering rates. Alternatively, if we picked a specific threshold, we would find that type 2 strains are associated with clustering (for lower thresholds), or that there are no differences in clustering among the two types (for higher thresholds). In any case, a classic molecular epidemiology analysis would conclude that type 2 is transmitting more than type 1, or that there are no differences in transmission between the two types. However, the shorter TBL and larger clustering rates of type 2 are due to its shorter latency and lower clock rate, not to increased transmission. Type 2 has a lower transmission rate compared to type 1, and it is bound to extinction, while type 1 is growing exponentially. This example shows the pitfalls of TBL and clustering analyses to study transmission in MTB epidemics. Admittedly, the simulation parameters for this example were picked to highlight the potential problems. However, the epidemiological characteristics of circulating MTB strains are normally not known, and the differences in latency and clock rates used here are possible. The length of the latent period and the clock rates are well within the range of values estimated with different data sets (*Menardo et al., 2019*; *Ku et al., 2021*). Overall, these results show that the standard interpretation of clustering results and TBL distributions can potentially be misleading in some epidemiological settings.

## Discussion
### The interpretation of clustering rates and TBL
Especially in low-incidence regions, clustering has proved to be useful to rapidly identify outbreaks and recent transmission (see *Walker et al., 2018* for an example). However, the use of clustering analyses and their interpretation has evolved with time, and went beyond the identification of linked bacterial strains. Researchers often look for association with clusters, or differences in clustering rates, to characterize the extent of transmission in different sub-populations coexisting in high-incidence areas. Recently, the distribution of TBL has been used in a similar fashion.

The results presented above demonstrate that these approaches suffer from two major limitations: (1) the transmission rate (per unit of time) does correlate positively with clustering rates, and negatively with TBL. However, it is not the only factor doing so. The lengths of the latency period, the molecular clock rate, the sampling period, and the sampling proportion all influence clustering

**Table 2.** Parameters and results for the different scenarios in the analysis of transmission dynamics.

$\lambda$: transmission rate, $\varepsilon$: sampling rate, R0: $\lambda/(\varepsilon+\sigma)$, $\sigma$: death rate, $\psi$: rate of progression to infectiousness, $\pi$: molecular clock rate in expected nucleotide changes per site per year, 95% SNP threshold: the minimum SNP threshold for which at least 95% of samples are clustered in at least 95% of simulations, 100% SNP threshold: the minimum SNP threshold for which 100% of samples are clustered in at least 95% of simulations, 95% CI TBL: the confidence interval for the overall TBL distribution, Mean TBL: average of the overall TBL distribution.

| Scenario (Median infectious period (months) - R0) | $\lambda$ | $\varepsilon$ | R0 | $\sigma$ | $\psi$ | $\pi$ | 95% SNP threshold | 100% SNP threshold | 95% CI TBL | Mean TBL |
|---|---|---|---|---|---|---|---|---|---|---|
| Long infectious period, shrinking (17–0.9) | 0.45 | 0.5 | 0.9 | 0 | 1 | $8 \times 10^{-8}$ | 7 | 16 | 0–3 | 0.62 |
| Long infectious period, stable (17 - 1) | 0.5 | 0.5 | 1 | 0 | 1 | $8 \times 10^{-8}$ | 6 | 16 | 0–3 | 0.59 |
| Long infectious period, growing (17–1.1) | 0.55 | 0.5 | 1.1 | 0 | 1 | $8 \times 10^{-8}$ | 6 | 16 | 0–3 | 0.57 |
| Medium infectious period, shrinking (8–0.9) | 0.9 | 1 | 0.9 | 0 | 1 | $8 \times 10^{-8}$ | 5 | 16 | 0–3 | 0.41 |
| Medium infectious period, stable (8 - 1) | 1 | 1 | 1 | 0 | 1 | $8 \times 10^{-8}$ | 5 | 15 | 0–2 | 0.38 |
| Medium infectious period, growing (8–1.1) | 1.1 | 1 | 1.1 | 0 | 1 | $8 \times 10^{-8}$ | 4 | 16 | 0–2 | 0.37 |
| Short infectious period, shrinking (4–0.9) | 1.8 | 2 | 0.9 | 0 | 1 | $8 \times 10^{-8}$ | 5 | 14 | 0–2 | 0.31 |
| Short infectious period, stable (4 - 1) | 2 | 2 | 1 | 0 | 1 | $8 \times 10^{-8}$ | 4 | 15 | 0–2 | 0.29 |
| Short infectious period, growing (4–1.1) | 2.2 | 2 | 1.1 | 0 | 1 | $8 \times 10^{-8}$ | 3 | 17 | 0–2 | 0.27 |

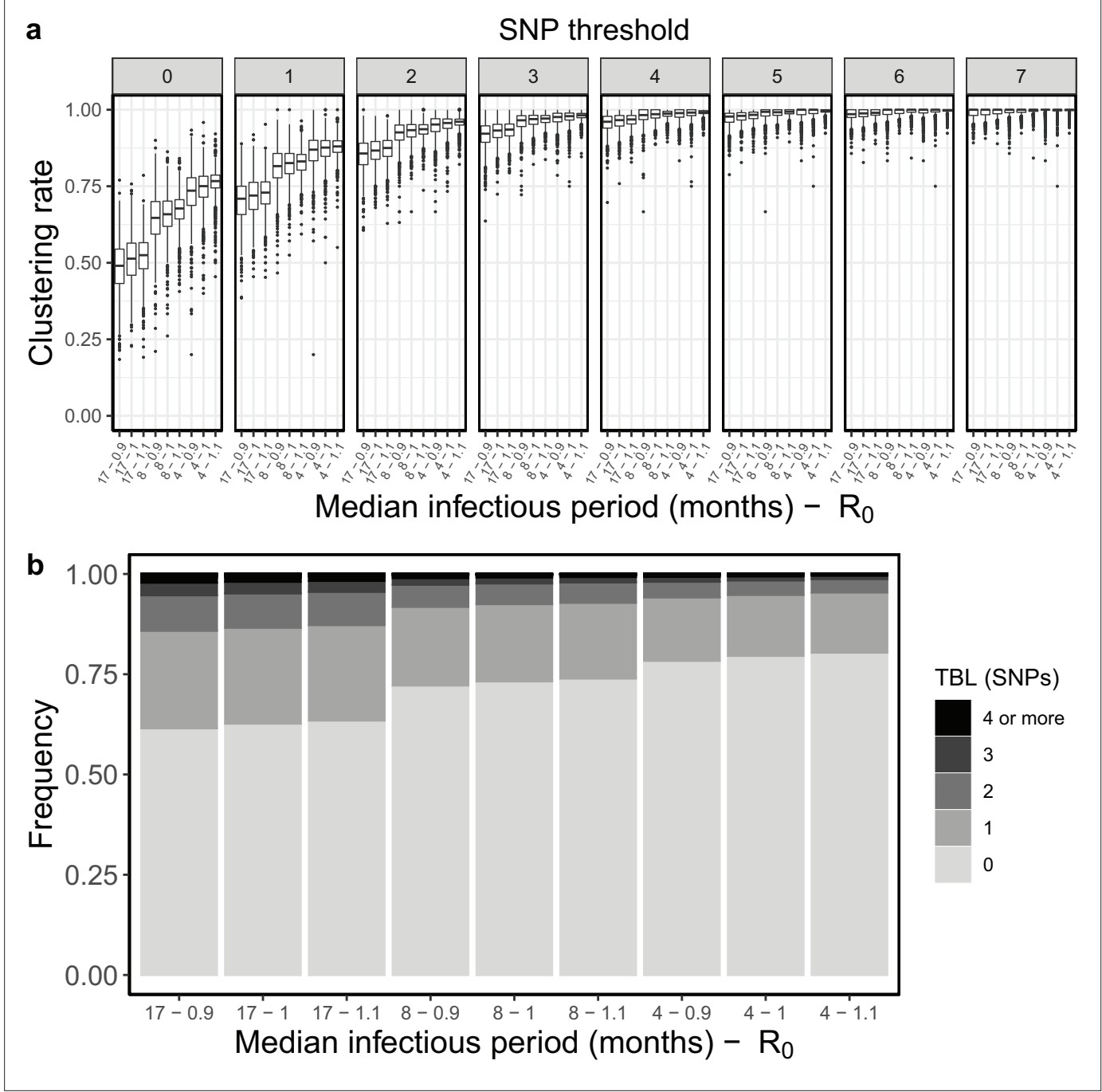

**Figure 3.** Clustering rates and TBL distributions for different transmission and sampling rates. (**a**) Clustering rates with different SNP thresholds. Only SNP thresholds up to the highest 95% sensitivity threshold are plotted (i.e. for thresholds higher than 7 SNPs more than 95% of samples are clustered in more than 95% of simulations for all settings). (**b**) Overall TBL distributions computed by merging all simulations.

and TBL. Therefore, differences between sub-populations might have nothing to do with differences in transmission. In other words, increased clustering might be due to shorter latency, lower clock rates, longer sampling periods, or higher sampling proportion, and not to increased transmission. (2) The transmission rate expressed per unit of time does not determine the dynamic of an epidemic. Whether an epidemic, or a sub-population is growing or not, depends on the ratio between the transmission rate ($\lambda$) and the rate at which infectious individuals stop being infectious ($\varepsilon+\sigma$). This ratio (R0) represents the average number of transmission events during an individual infectious period

**Table 3.** Parameters and results for the two simulated scenarios in the practical example.

$\lambda$: transmission rate, $\varepsilon$: sampling rate, $R_0 = \lambda/(\varepsilon+\sigma)$, $\sigma$: death rate, $\psi$: rate of progression to infectiousness, $\pi$: molecular clock rate in expected nucleotide changes per site per year, 95% SNP threshold: the minimum SNP threshold for which at least 95% of samples are clustered in at least 95% of simulations, 100% SNP threshold: the minimum SNP threshold for which 100% of samples are clustered in at least 95% of simulations, 95% CI TBL: the confidence interval for the overall TBL distribution, Mean TBL: average of the overall TBL distribution.

| Scenario | $\lambda$ | $\varepsilon$ | $R_0$ | $\sigma$ | $\psi$ | $\pi$ | 95% SNP threshold | 100% SNP threshold | 95% CI TBL | Mean TBL |
|---|---|---|---|---|---|---|---|---|---|---|
| Type 1 | 1.1 | 0.5 | 1.1 | 0.5 | 0.7 | $1 \times 10^{-7}$ | 8 | 21 | 0–4 | 0.82 |
| Type 2 | 0.9 | 0.5 | 0.9 | 0.5 | 1.7 | $7 \times 10^{-8}$ | 5 | 13 | 0–2 | 0.40 |

(transmission per generation). Unless all other parameters are held fixed, R0 does not correlate with clustering rates or TBL. Consequently, TBL and clustering rates can be used to estimate whether a sub-population is stable, shrinking, or growing (in relative terms compared to another sub-population), only by assuming that the clock rate, the latency and infectious periods, the sampling period, and the sampling proportion are identical in the two sub-populations. Moreover, the simple comparison of clustering rates and TBL distributions without formal statistical inference disregards stochastic effects. The same epidemiological process can generate different clustering rates and TBL. Statistical inference is necessary to test whether the observed differences are significant. Altogether, these results resonate with the findings of **Poon, 2016**, who reported that clustering methods used to study the epidemiology of HIV were biased towards detecting different sampling rates among sub-populations, and not variation in transmission rates.

In addition to clustering rates and TBL, researchers can use complementary data to understand the epidemiological dynamics. For example, the change through time in the proportion of cases belonging to a certain type can be used as a proxy for the relative reproductive number (i.e. if a type is increasing in proportion within a population, it should have a larger R0 compared to other types). Similarly, if the absolute number of cases is increasing, R0 should be greater than one. However, there can be confounding factors, such as the migration of strains from other regions. Finally, the length of the infectious and latent period can be estimated (with some assumptions) from prevalence surveys and notification data (**Ku et al., 2021**). Usually this is done at the population level, the same analysis on individual lineages, or clones, could provide useful insights on the heterogeneity of MTB populations.

## Is there an optimal SNP threshold?

The optimal SNP threshold is the one that maximizes sensitivity and specificity. Here I defined the 95% sensitivity SNP threshold as the minimum threshold for which at least 95% of strains are clustered in at least 95% of the simulations (Materials and methods). In other words, this is the threshold that maximizes specificity at a 95% sensitivity level. One important result is that this threshold depends strongly on the epidemiological conditions and on the sample size: across all scenarios simulated for this study the 95% sensitivity threshold ranged between 3 and 11 SNPs, and more extreme values are not impossible. Ideally, molecular epidemiological studies should use larger thresholds for settings characterized by longer latency, lower transmission rates, shorter sampling periods, and/or lower sampling proportions. However, if a MTB population is not uniform, but consists of sub-populations with different epidemiological characteristics, or molecular clock rates, using a single SNP threshold will lead to biased results.

## Biology and epidemiology of MTB lineages

The issues discussed above are most relevant when comparing different bacterial sub-populations, such as the MTB lineages. Different MTB lineages have different clustering rates and distributions of TBL, independently from the region of sampling. For example, compared to other lineages, L2 was consistently found to have shorter TBL and higher clustering rates virtually everywhere, including in Vietnam (**Holt et al., 2018**, **Hang et al., 2019**), Malawi (**Guerra-Assunção et al., 2015**; **Sobkowiak et al., 2020**), Uzbekistan (**Merker et al., 2018**), South Africa (**Cox et al., 2021**), Georgia (**Gygli et al.,**

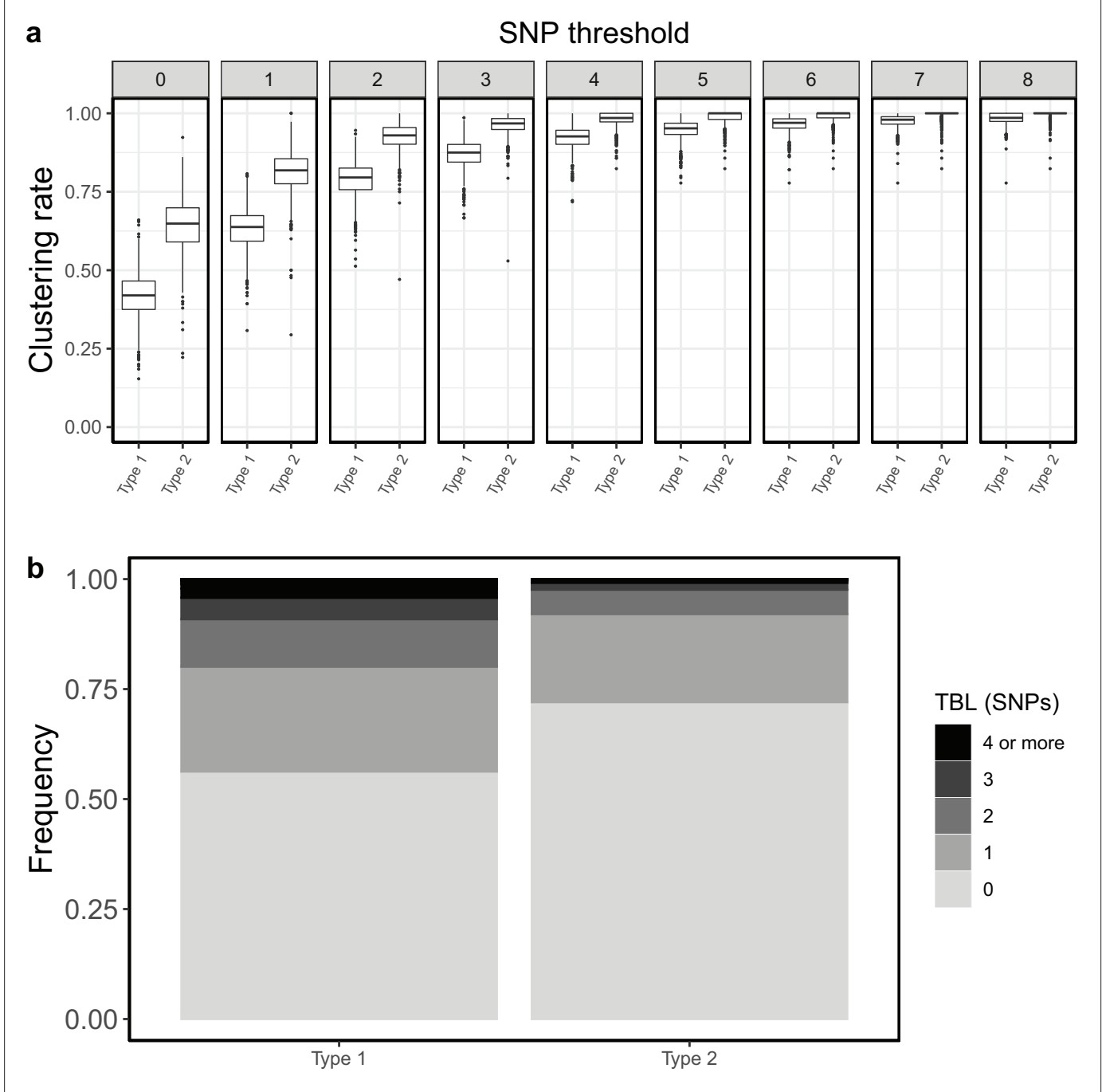

**Figure 4.** Clustering rates and TBL distributions for two different hypothetical sub-populations. Type 1 is expanding (R0=1.1), it has a long latency period (median:~12 months), and a clock rate of $1 \times 10^{-7}$. Type 2 has a R0=0.9, a short latency period (median:~5 months), and a clock rate of $7 \times 10^{-8}$. (**a**) Clustering rates for the two types with different SNP thresholds. Only SNP thresholds up to the highest 95% sensitivity threshold are plotted (i.e. for higher thresholds more than 95% of samples are clustered in more than 95% of simulations for all settings). (**b**) Overall TBL distributions computed by merging all simulations.

2021), Iran (**Vaziri et al., 2019**), as well as globally (**Freschi et al., 2021**). At the opposite end of the spectrum, L1 was repeatedly found to have longer terminal branches, and lower clustering rates, compared to other lineages (**Guerra-Assunção et al., 2015**; **Holt et al., 2018**, **Hang et al., 2019**, **Sobkowiak et al., 2020**; **Freschi et al., 2021**). This repeated pattern is probably due to intrinsic bacterial factors, which affect clustering and TBL in all epidemiological settings. Different sampling proportions and sampling periods among lineages are unlikely to be responsible for this widespread

pattern. The three remaining factors that could explain the global differences between L2 and L1 are: (1) clock rates, (2) latency, and (3) transmission rates per unit of time.

1. A faster molecular clock for L1 would explain the lower clustering rates and longer terminal branches. However, the little evidence that is available does not support this hypothesis. L1 and L2 were found to have similar clock rates when analyzed with the same set of methods, although the uncertainty of the estimates was very large (*Menardo et al., 2019*). Among other studies that inferred the clock rate for L2, most confirmed a moderately high rate (~1 × 10⁻⁷, *Merker et al., 2018*; *Rutaihwa et al., 2019*; *Torres Ortiz et al., 2021*), while one estimated a higher rate of evolution (~3 × 10⁻⁷, *Eldholm et al., 2016*). The only additional study to attempt the estimation of the clock rate of L1 found a lack of a temporal signal (*Menardo et al., 2021*). Altogether the available evidence is limited, and more precise estimates are needed to understand whether the rate of molecular evolution contributes to the different clustering rates and TBL for L1 and L2.

2. Longer latency would cause low clustering rates and long terminal branches in L1. Here, latency is defined as the period between being infected and becoming infectious, and it is typically thought to correspond to the period of asymptomatic infection. Although, pre-symptomatic transmission could be more important than previously thought (*Kendall et al., 2021*). There is some indirect evidence for a longer asymptomatic period in L1: a recent study estimated that asymptomatic infection is longer in TB patients in Southeast Asian countries such as Lao, Cambodia, and the Philippines (*Ku et al., 2021*), where L1 is responsible for more than half of TB cases (*Schopfer et al., 2015*; *Chen et al., 2017*; *Somphavong, 2018*; *Netikul et al., 2021*). In these countries, latency was found to be up to three times longer compared to Pakistan, Ethiopia and Zambia, where L1 is rare (*Mulenga et al., 2010*; *Firdessa et al., 2013*; *Chihota et al., 2018*; *Tulu and Ameni, 2018*; *Wiens et al., 2018*; *Ali et al., 2019*). Moreover, it is well documented that L2 is the most virulent MTB lineage (*Hanekom et al., 2011*; *Peters et al., 2020*), while L1 is less virulent compared to L2, L3, and L4 (*Bottai et al., 2020*). Virulence is often measured as bacterial growth rate in animal models or in human cell lines, and it seems quite natural that an infection of faster growing bacteria would have a shorter latency period.

3. A higher transmission rate per unit of time would also cause higher clustering rates and lower terminal branches in L2. This is how the results of these analyses are typically interpreted. Some partial evidence for this comes from a study in The Gambia, that found that household contacts of patients infected with L2 strains were more likely to develop disease within two years, compared to contacts of patients infected with other strains (*de Jong et al., 2008*). This could be the result of shorter latency, or higher transmission rates for L2. If L2 has a constitutively higher transmission rate, its proportion in a MTB population is expected to be stable only if the infectious period is shorter compared to other lineages. For example, in Malawi L2 was found to have higher clustering rates compared to L1, however the proportion of cases caused by the two lineages did not change over 20 years of monitoring (*Sobkowiak et al., 2020*). Assuming higher transmission rates for L2, these results can only be explained with a shorter infectious period of L2 compared to L1. Similarly to latency, the period between onset of symptoms and diagnosis was estimated to be longer in Southeast Asian countries where the MTB populations are dominated by L1 (*Ku et al., 2021*). This trend was not as strong as in the case of latency, but this is not surprising, as the delay in seeking care and receiving treatment does not depend only on bacterial and host factors, but also on the public health system and other social conditions. As for latency, higher virulence could explain the higher transmission rate and shorter infectious period for L2 compared to L1.

To summarize, different clustering results and TBL between L1 and L2 are likely caused by differences in latency and/or transmission rate per unit of time. However, the analysis of TBL distribution and clusters cannot tease these two factors apart. In this discussion I focused on L1 and L2, as these two lineages represent the extremes in terms of clustering and TBL. For all other lineages the same logic applies, differences in latency, transmission, and clock rates influence the tree topology in different combinations, resulting in intermediate TBL and clustering rates. Importantly, the effects of all these factors are additive. For example, in a lineage with longer latency and lower clock rate, the longer latency would result in lower clustering rates and longer TBL, while the lower clock rate has the opposite effect. The outcome will be determined by the magnitude of the changes: if the variation in clock rates is large enough to compensate for the longer latency, clustering rates will be larger and TBL shorter, otherwise clustering rates will be lower, and TBL longer.

## Conclusions

The take home message of this study is that clustering analyses and TBL can tell us only so much about the dynamics of MTB epidemics. While clustering will continue to be useful to detect linked strains, conclusions about differences in transmission among sub-populations are at best a simplification that conflates many different factors, at worst outright wrong. Phylodynamic methods that estimate the parameters of an epidemiological model from genomic data are becoming available (*Kühnert et al., 2016*; *Didelot et al., 2017*; *Volz and Siveroni, 2018*). However, these analyses have limits that cannot be overcome exclusively with phylogenetic data (*Louca et al., 2021*), and in any case they are challenging with current MTB data sets (*Kühnert et al., 2018*; *Walter et al., 2022*). Methodological advances, the integration of different types of data, and more complete and longer sampling series of MTB epidemics will allow to study epidemiological dynamics more accurately in the future. In the meantime, the results of clustering and TBL analyses should not be over-interpreted.

# Materials and methods

## Simulation pipeline

MTB epidemics were simulated using an Exposed-Infectious epidemiological model. The model has two compartments, one for infectious individuals (I) and one for exposed individuals (E), the latter contains individuals that have been infected but are not yet infectious. Individuals in compartment I generate new infections adding new exposed individuals in compartment E with rate $\lambda$. Exposed individuals become infectious moving from compartment E to compartment I with rate $\psi$. Individuals are removed from compartment I either by death (or self-cure), occurring at rate σ or sampling, occurring at rate $\varepsilon$. Under this model the reproductive number R0 corresponds to $\lambda/(\sigma+\varepsilon)$.

Each of these events is modeled as a Poisson process, so that the probability of the event to occur through time is exponentially distributed. The mean and median of the exponential distribution are calculated respectively as 1/rate and ln(2)/rate. For example, with $\psi$=1, the average and median latency periods are 1 and ~0.69 years, respectively.

The BEAST v2.6.6 (*Bouckaert et al., 2019*) package MASTER v6.1.2 (*Vaughan and Drummond, 2013*) was used to simulate phylogenetic trees under the parameters of the epidemiological model. Simulations were stopped after 30 years, or before in case the simulated lineage went extinct.

A post-simulation condition was implemented by specifying the minimum and maximum number of tips in the phylogenetic tree necessary to accept the simulation. This threshold regard the total number of tips sampled during the simulated period, not the number of lineages existing at the most recent time point. Unless differently specified in the text, the minimum number of tips for the simulated tree was 100, the maximum was 2500. Because different sampling periods can result in different TBL and clustering rates (see Results), for the analysis of transmission rates, infectious period, R0, and for the example, an additional post-simulation condition on the tree height was set: the simulation was accepted only if the tree height was at least 10 years. If the simulation was accepted, the tree was subsampled to the most recent 10 years of sampling, that is, all tips sampled more than 10 years before the most recent tip were discarded. In some analysis, a different sampling period was used, this was specified in the text. After this step, an outgroup was added to the phylogenetic tree, so that the tree could be rooted for downstream analyses.

Seq-gen v1.3.4 (*Rambaut and Grass, 1997*) was used to simulate genome sequences given the phylogenetic trees and a clock rate in expected nucleotide changes per site per year. One chromosome of 4 Mb was simulated for each tip, corresponding to the size of the MTB genome that is usually considered after discarding repetitive regions (*Brites et al., 2018*). Simulations were run under a HKY+$\Gamma$ model with transition-transversion ratio k=2 and shape parameter for the gamma distribution $\alpha$=1.

Variable sites were extracted from the whole genome alignments with SNP-sites v2.5.1 (*Page et al., 2016*), recording the number of invariant sites. Phylogenetic trees were inferred form the SNP alignment with RAxML-NG v0.9.0 (*Kozlov et al., 2019*) using a HKY model, and 2 starting trees (1 random, 1 parsimony). The maximum likelihood tree was rooted using the outgroup, and the branch lengths were converted in expected number of SNPs multiplying them by the alignment length (the number of SNPs). Finally, Treecluster v1.0.3 (*Balaban et al., 2019*) was used to obtain clusters under

different SNP thresholds $t$ = (0, 1... 49, 50), so that the maximum pairwise distance between tips in the cluster was at most $t$ (method 'max').

1000 simulations were performed for all tested conditions. Clustering rates were computed for each simulation individually, while the TBL distributions were computed merging all simulations with the same set of parameters.

The pipeline is wrapped in a python script, using Biopython v1.78 (*Cock et al., 2009*) and ETE3 v3.1.2 (*Huerta-Cepas et al., 2016*) to handle sequences and trees. Plots were generated with the R package ggplot2 v3.1.1 (*Wickham, 2011*).

The simulation pipeline, scripts and data to reproduce these results are available at https://github.com/fmenardo/sim_cluster_MTB, (copy archived at swh:1:rev:aa2e0bb7629c46e64a099247d225466615b55b07, *Menardo, 2022*).

### Rationale for the 95% and 100% SNP thresholds

Under the model used in this study all simulated samples are, by definition, part of a single transmission chain. Therefore, the ground truth is assumed to be that all samples should be clustered. Under these conditions, the sensitivity of an SNP threshold can be estimated directly, it is the percentage of samples that are clustered. The specificity, however, cannot be estimated, as it depends on many other factors that are not included in the model (e.g. migration).

The two thresholds that are reported in the results, that is, the minimum SNP threshold at which at least 95% (or 100%) of samples are clustered in 95% of simulations, maximize specificity, at known acceptable sensitivity levels. These two thresholds are useful because they show that the specificity and sensitivity of a threshold depend on the epidemiological conditions and other factors such as clock rates and sampling (see Results).

## Acknowledgements

This work was founded by the SNF Ambizione grant PZ00P3_193473. Computations were performed on the ScienceCluster at the University of Zurich.

## Additional information

### Funding

| Funder | Grant reference number | Author |
| --- | --- | --- |
| Schweizerischer Nationalfonds zur Förderung der Wissenschaftlichen Forschung | PZ00P3_193473 | Fabrizio Menardo |

The funders had no role in study design, data collection and interpretation, or the decision to submit the work for publication.

### Author contributions

Fabrizio Menardo, Conceptualization, Formal analysis, Funding acquisition, Investigation, Methodology, Software, Visualization, Writing - original draft, Writing - review and editing

### Author ORCIDs

Fabrizio Menardo ⓘ http://orcid.org/0000-0002-7885-4482

### Decision letter and Author response

Decision letter https://doi.org/10.7554/eLife.76780.sa1
Author response https://doi.org/10.7554/eLife.76780.sa2

## Additional files

### Supplementary files
• Transparent reporting form

### Data availability
The current manuscript is a computational study, so no data have been generated for this manuscript. Modelling code and simulation results are available at https://github.com/fmenardo/sim_cluster_MTB, (copy archived at swh:1:rev:aa2e0bb7629c46e64a099247d225466615b55b07).

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

## Appendix 1

### Molecular clock rate

It was pointed out previously that faster evolving lineages will accumulate more mutations, and therefore have longer TBL and lower clustering rates (*Stimson et al., 2019*), although this is often overlooked. I tested three different rates of molecular evolution ($4 \times 10^{-8}$, $8 \times 10^{-8}$ and $1.2 \times 10^{-7}$ nucleotide changes per site per year), roughly corresponding to the range of possible clock rates in MTB (*Menardo et al., 2019*). All other parameters were identical in all simulations ($\lambda = \psi=1$; $\sigma = \varepsilon=0.5$, sampling time = 10 years). For this analysis, I simulated 1,000 transmission trees, I then simulated the molecular evolution with different clock rates on the same set of trees. As expected, clustering rates were higher with low clock rates (*Appendix 1—figure 1*), with the 95% sensitivity threshold equal to 4, 6 and 9, respectively for the lowest, mid, and highest clock rate. Also, the TBL distributions were markedly different, with longer terminal branches for faster clock rates (*Appendix 1—table 1*).

**Appendix 1—table 1.** Parameters and results for the different simulated scenarios in the analysis of clock rates.

$\lambda$: transmission rate, $\varepsilon$: sampling rate, R0 = $\lambda /(\varepsilon+\sigma)$, $\sigma$: death rate, $\psi$: rate of progression to infectiousness, $\pi$: molecular clock rate in expected nucleotide changes per site per year, 95% SNP threshold: the minimum SNP threshold for which at least 95% of samples are clustered in at least 95% of simulations, 100% SNP threshold: the minimum SNP threshold for which 100% of samples are clustered in at least 95% of simulations, 95% CI TBL: the confidence interval for the overall TBL distribution, Mean TBL: average of the overall TBL distribution.

| Scenario | $\lambda$ | $\varepsilon$ | R0 | $\sigma$ | $\psi$ | $\pi$ | 95% SNP threshold | 100% SNP threshold | 95% CI TBL | Mean TBL |
|---|---|---|---|---|---|---|---|---|---|---|
| Fast clock rate | 1 | 0.5 | 1 | 0.5 | 1 | $1.2 \times 10^{-7}$ | 9 | 24 | 0–4 | 0.85 |
| Mid clock rate | 1 | 0.5 | 1 | 0.5 | 1 | $8 \times 10^{-8}$ | 6 | 17 | 0–3 | 0.56 |
| Low clock rate | 1 | 0.5 | 1 | 0.5 | 1 | $4 \times 10^{-8}$ | 4 | 9 | 0–2 | 0.28 |

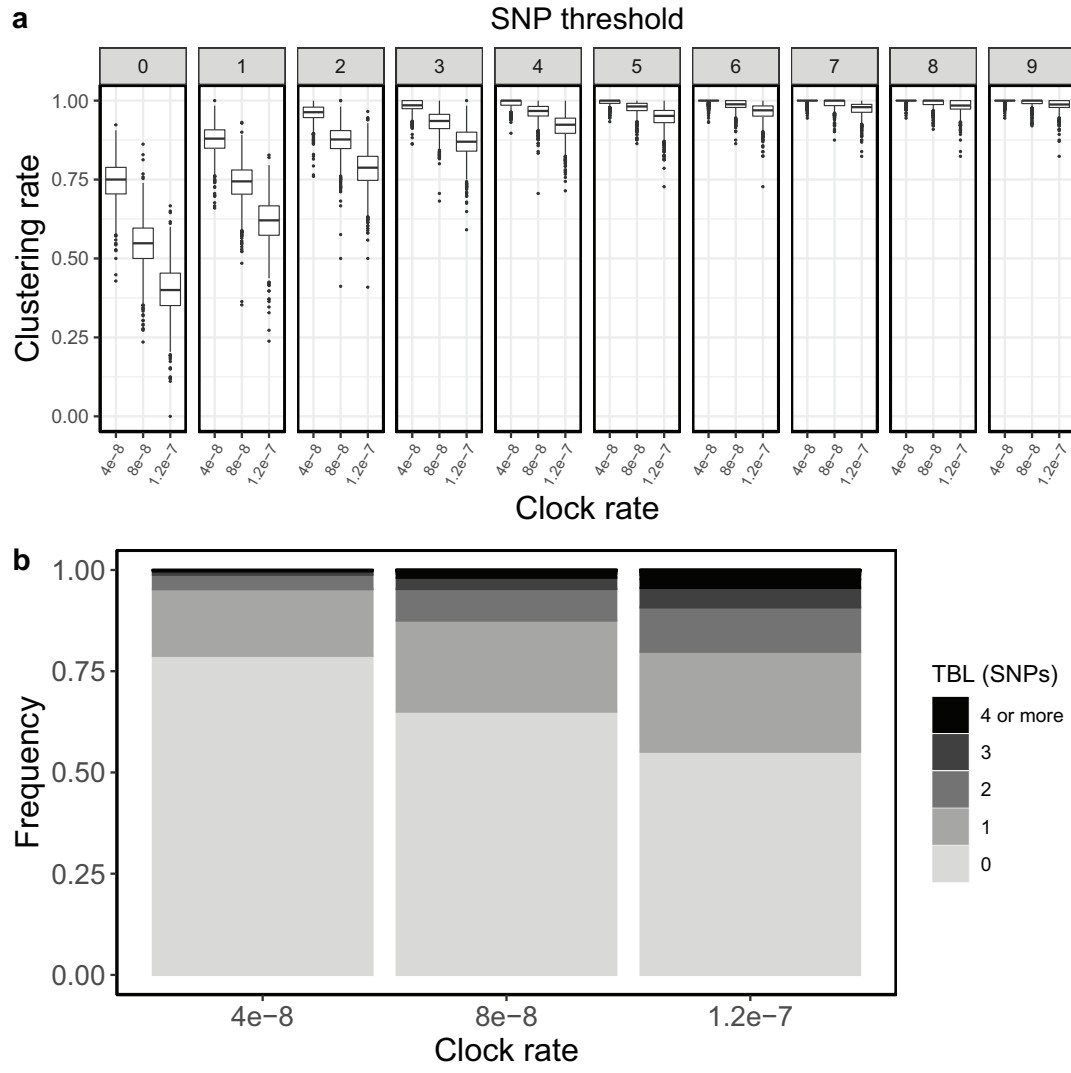

**Appendix 1—figure 1.** Clustering rates and TBL distributions for different molecular clock rates. (**a**) Clustering rates with different SNP thresholds. Only SNP thresholds up to the highest 95% sensitivity threshold are plotted (i.e. for higher thresholds more than 95% of samples are clustered in more than 95% of simulations for all settings). (**b**) Overall TBL distributions computed by merging all simulations.

# Appendix 2

## Sampling proportion

To test how different sampling proportions affect clustering rates and TBL, I used 4 different combinations of $\sigma$ and $\varepsilon$ (0.75–0.25; 0.5–0.5; 0.25–0.75; 0–1), corresponding to 4 sampling proportions, in which 25%, 50%, 75%, and 100% of all cases are sampled. All other parameters were kept constant ($\pi$=8 × 10$^{-8}$, $\lambda$ = $\psi$=1, sampling time = 10 years). Lower sampling proportions resulted in lower clustering rates and longer terminal branch lengths (***Appendix 2—figure 1***). The 95% sensitivity threshold was equal to 8, 6, 5, 4 for a sampling proportion of 25%, 50–75%, and 100% respectively, while the mean of the TBL distribution was 0.83, 0.56, 0.45, and 0.39 (***Appendix 2—table 1***).

**Appendix 2—table 1.** Parameters and results for the different simulated scenarios in the analysis of sampling proportions.

$\lambda$ : transmission rate, $\varepsilon$: sampling rate, R0 = $\lambda$ /($\varepsilon$+$\sigma$), $\sigma$: death rate, $\psi$: rate of progression to infectiousness, $\pi$: molecular clock rate in expected nucleotide changes per site per year, 95% SNP threshold: the minimum SNP threshold for which at least 95% of samples are clustered in at least 95% of simulations, 100% SNP threshold: the minimum SNP threshold for which 100% of samples are clustered in at least 95% of simulations, 95% CI TBL: the confidence interval for the overall TBL distribution, Mean TBL: average of the overall TBL distribution.

| Scenario | $\lambda$ | $\varepsilon$ | R0 | $\sigma$ | $\psi$ | $\pi$ | 95% SNP threshold | 100% SNP threshold | 95% CI TBL | Mean TBL |
|---|---|---|---|---|---|---|---|---|---|---|
| 25% sampling proportion | 1 | 0.25 | 1 | 0.75 | 1 | 8 × 10$^{-8}$ | 8 | 19 | 0–4 | 0.83 |
| 50% sampling proportion | 1 | 0.5 | 1 | 0.5 | 1 | 8 × 10$^{-8}$ | 6 | 17 | 0–3 | 0.56 |
| 75% sampling proportion | 1 | 0.75 | 1 | 0.25 | 1 | 8 × 10$^{-8}$ | 5 | 16 | 0–3 | 0.45 |
| 100% sampling proportion | 1 | 1 | 1 | 0 | 1 | 8 × 10$^{-8}$ | 4 | 16 | 0–2 | 0.39 |

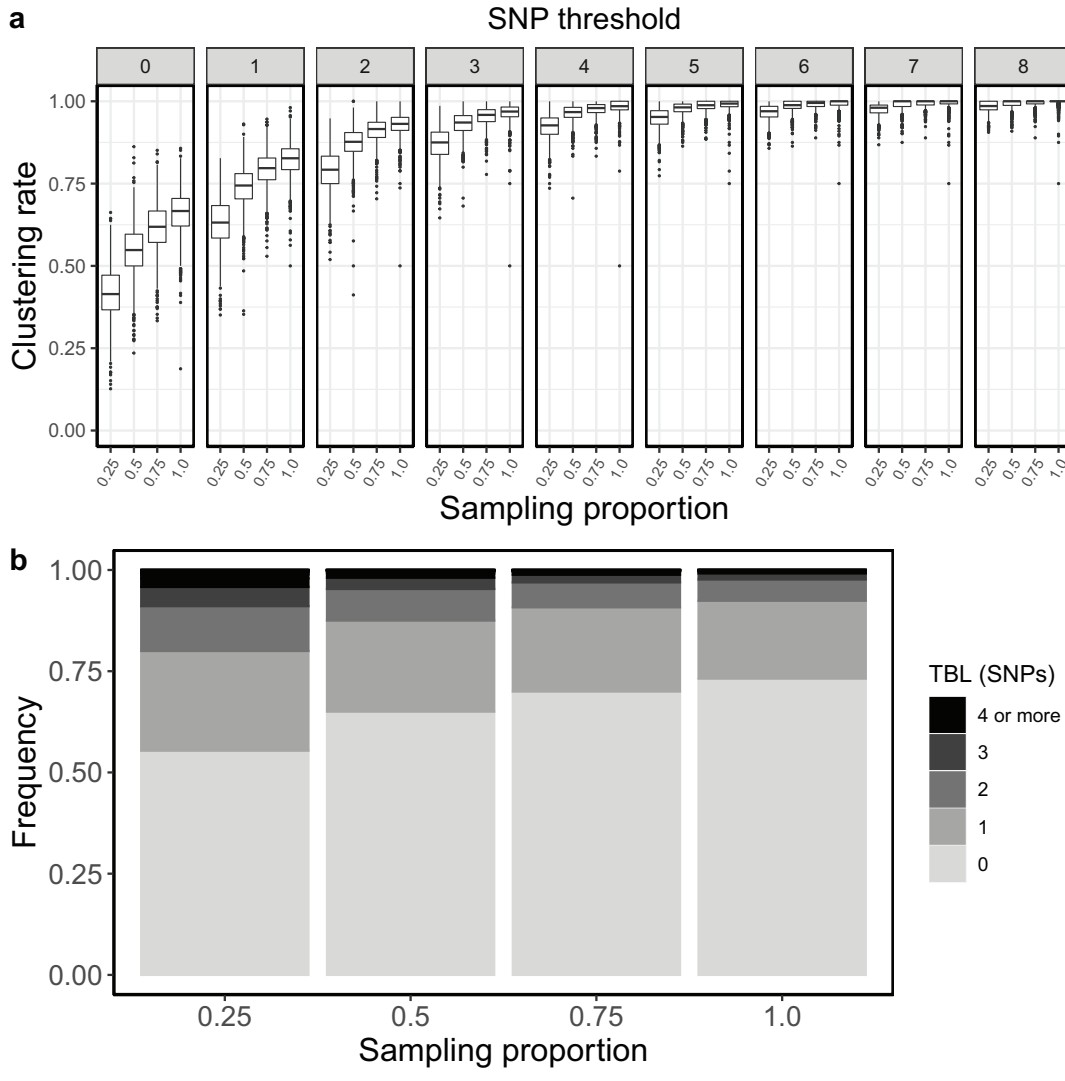

**Appendix 2—figure 1.** Clustering rates and TBL distribution for different sampling proportions. (**a**) Clustering rates with different SNP thresholds. Only SNP thresholds up to the highest 95% sensitivity threshold are plotted (i.e. for higher thresholds more than 95% of samples are clustered in more than 95% of simulations for all settings). (**b**) Overall TBL distributions computed by merging all simulations.

# Appendix 3

## Sampling period

To test whether the length of the sampling period could affect clustering rates and TBL I simulated three different scenarios in which the only difference was that the epidemic was sampled for 5, 10, and 20 years, respectively. All other parameters were kept fixed ($\pi = 8 \times 10^{-8}$, $\lambda = \psi = \varepsilon = 1$, $\sigma = 0$). I found that shorter sampling periods resulted in lower clustering rates and longer terminal branches (*Appendix 3—figure 1* and *Appendix 3—table 1*). This is probably because with shorter sampling periods the proportion of samples at the edges of the sampling window is larger. These samples are less likely to be clustered, as only one side of the transmission chain is present in the sample set.

**Appendix 3—table 1.** Parameters and results for the different simulated scenarios in the analysis of different sampling periods.

$\lambda$: transmission rate, $\varepsilon$: sampling rate, R0 = $\lambda /(\varepsilon + \sigma)$, $\sigma$: death rate, $\psi$: rate of progression to infectiousness, $\pi$: molecular clock rate in expected nucleotide changes per site per year, 95% SNP threshold: the minimum SNP threshold for which at least 95% of samples are clustered in at least 95% of simulations, 100% SNP threshold: the minimum SNP threshold for which 100% of samples are clustered in at least 95% of simulations, 95% CI TBL: the confidence interval for the overall TBL distribution, Mean TBL: average of the overall TBL distribution.

| Scenario | $\lambda$ | $\varepsilon$ | R0 | $\sigma$ | $\psi$ | $\pi$ | 95% SNP threshold | 100% SNP threshold | 95% CI TBL | Mean TBL |
|---|---|---|---|---|---|---|---|---|---|---|
| 5 years sampling | 1 | 0.5 | 1 | 0 | 1 | $8 \times 10^{-8}$ | 9 | 18 | 0–3 | 0.47 |
| 10 years sampling | 1 | 1 | 1 | 0 | 1 | $8 \times 10^{-8}$ | 4 | 13 | 0–2 | 0.39 |
| 20 years sampling | 1 | 0.5 | 1 | 0 | 1 | $8 \times 10^{-8}$ | 3 | 11 | 0–2 | 0.34 |

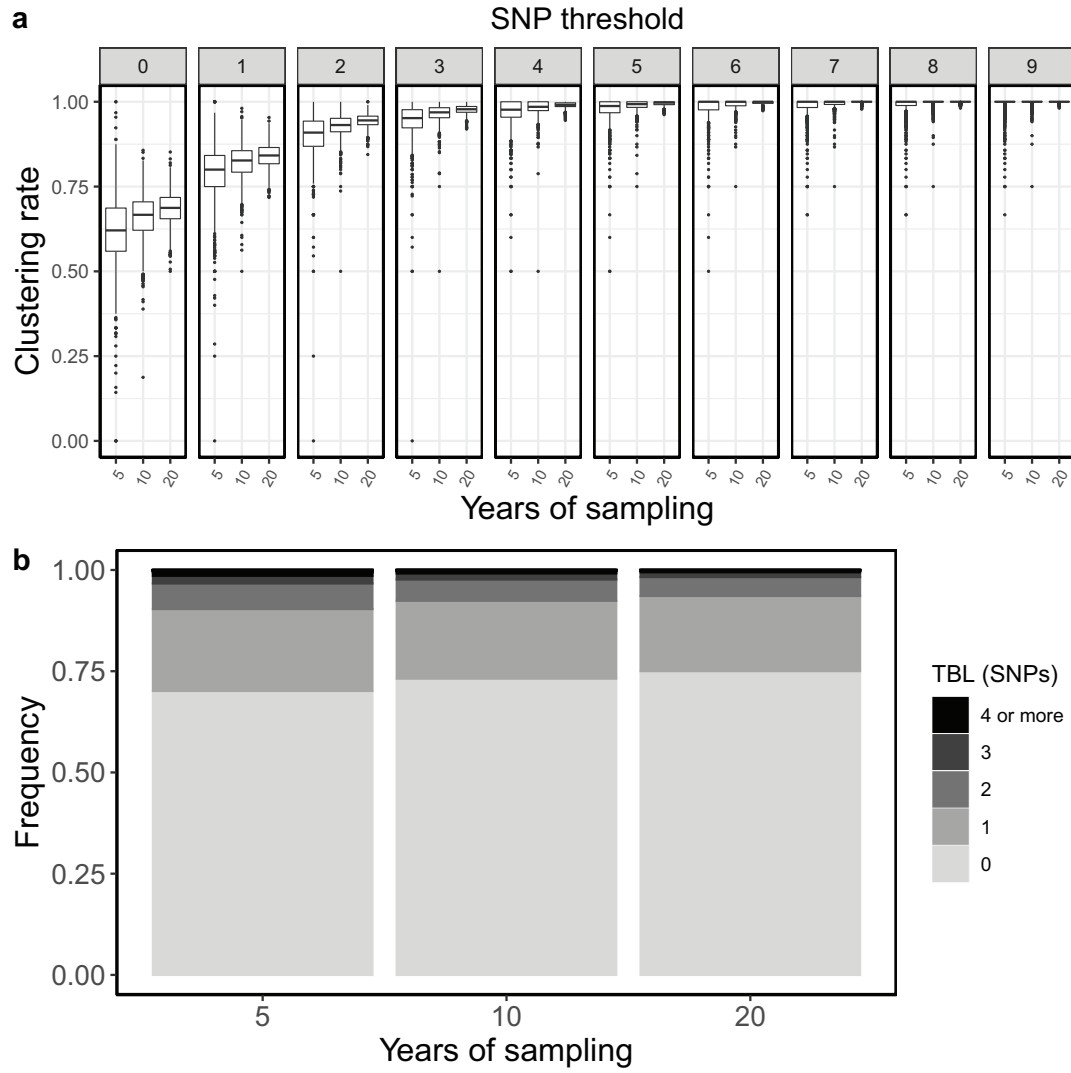

**Appendix 3—figure 1.** Clustering rates and TBL distributions for different sampling periods. (**a**) Clustering rates with different SNP thresholds. Only SNP thresholds up to the highest 95% sensitivity threshold are plotted (i.e. for higher thresholds more than 95% of samples are clustered in more than 95% of simulations for all settings). (**b**) Overall TBL distributions computed by merging all simulations.

## Appendix 4

### Minimum tree size

All results presented so far are based on the same simulation strategy: lineages are evolved for up to 30 years, or until they go extinct, and the last 10 years are sampled (except for the analysis of the sampling period, in which I tested different values, see above). The only condition imposed on a simulation to be accepted is a minimum and maximum number of tips in the transmission tree simulated by MASTER (before sub-sampling the last 10 years). The maximum tip number was set to 2500, which is seldom reached within 30 years of simulated evolution, while the minimum number of tips was set to 100. I tested whether a smaller minimum number of tips could change the clustering rates and TBL. I used the same settings used in the mid clock rate analysis presented above, the only difference was that the minimum number of tips was set to 25, 50, or 100. The TBL distribution showed only minor differences (*Appendix 4—table 1*). The median of the clustering rates was also similar for all settings, the only difference was that the distribution of clustering rates was more dispersed with a lower number of tips (*Appendix 4—figure 1*). This caused the 95% sensitivity thresholds to be different: 11, 8, and 6 SNPs, respectively with 25, 50, and 100 minimum tips. The larger dispersion was caused by the lower sample sizes, indeed the minimum threshold for which 100% of the samples were clustered in at least 95% of the simulations showed no trend (16, 17, and 17 SNPs from the lowest to the highest threshold).

**Appendix 4—table 1.** Parameters and results for the different simulated scenarios in the analysis of the minimum number of tips to accept a simulation.

$\lambda$: transmission rate, $\varepsilon$: sampling rate, R0 = $\lambda /(\varepsilon+\sigma)$, $\sigma$: death rate, $\psi$: rate of progression to infectiousness, $\pi$: molecular clock rate in expected nucleotide changes per site per year, 95% SNP threshold: the minimum SNP threshold for which at least 95% of samples are clustered in at least 95% of simulations, 100% SNP threshold: the minimum SNP threshold for which 100% of samples are clustered in at least 95% of simulations, 95% CI TBL: the confidence interval for the overall TBL distribution, Mean TBL: average of the overall TBL distribution.

| Scenario | $\lambda$ | $\varepsilon$ | R0 | $\sigma$ | $\psi$ | $\pi$ | 95% SNP threshold | 100% SNP threshold | 95% CI TBL | Mean TBL |
|---|---|---|---|---|---|---|---|---|---|---|
| Min tips = 25 | 1 | 0.5 | 1 | 0.5 | 1 | $8 \times 10^{-8}$ | 11 | 16 | 0–3 | 0.59 |
| Min tips = 50 | 1 | 0.5 | 1 | 0.5 | 1 | $8 \times 10^{-8}$ | 8 | 17 | 0–3 | 0.57 |
| Min tips = 100 | 1 | 0.5 | 1 | 0.5 | 1 | $8 \times 10^{-8}$ | 6 | 17 | 0–3 | 0.56 |

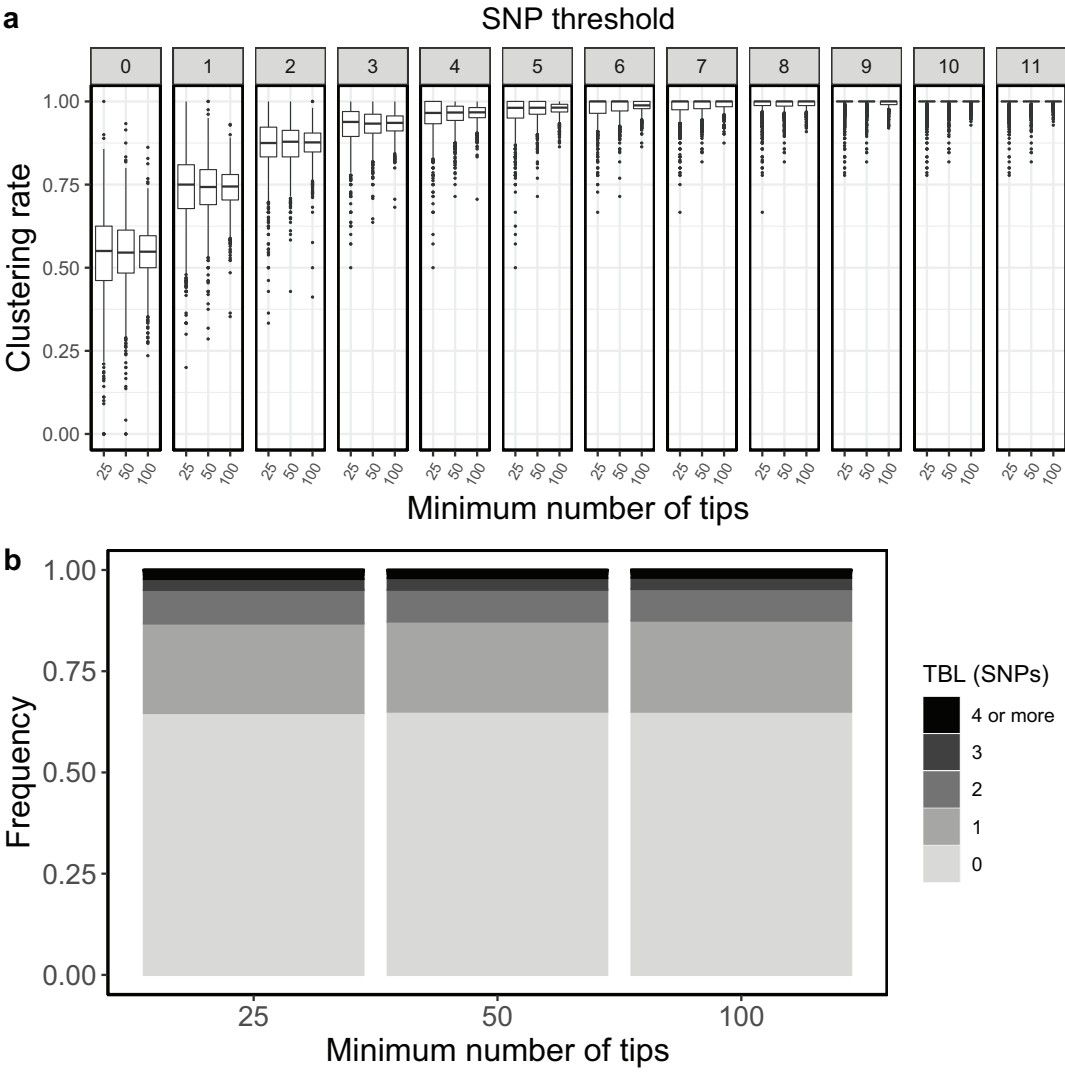

**Appendix 4—figure 1.** Clustering rates and TBL distributions for different minimum number of tips necessary to accept the MASTER simulation. (**a**) Clustering rates with different SNP thresholds. Only SNP thresholds up to the highest 95% sensitivity threshold are plotted (i.e. for higher thresholds more than 95% of samples are clustered in more than 95% of simulations for all settings). (**b**) Overall TBL distributions computed by merging all simulations.

## Appendix 5

### Transmission and sampling rates

I tested how the transmission and sampling rates affect the results of clustering and the TBL distribution. I used five different values for the transmission rate ($\lambda$ =0.8, 0.9, 1, 1.1, 1.2), keeping all other parameters constant ($\pi$=8 × 10$^{-8}$, $\sigma$=0, $\varepsilon$=1, $\psi$=1, sampling time = 10 years). For these analyses I excluded all simulations in which the lineage went extinct in the first ten years, so that the sampling period of the different scenarios is similar (see Materials and methods). I found that higher transmission rates resulted in larger clustering rates and shorter TBL (*Appendix 5—figure 1*, *Appendix 5—table 1*). The 95% sensitivity threshold ranged from 6 SNPs, for $\lambda$ =0.8,–4 SNPs for $\lambda$ =1.2; while the mean of the TBL distribution ranged from 0.43 to 0.34, for $\lambda$ =0.8 and 1.2, respectively. Next, I considered different lengths of the infectious period, to do this I used a constant transmission rate $\lambda$ =1, and five different values of the sampling rate ($\varepsilon$=1.25,1.11111, 1, 0.90909, 0.83333), corresponding to different median infectious periods, included in the range between 6.9 and 10.4 months. I found essentially no differences in the clustering rates for different lengths of the infectious period (*Appendix 5—figure 2*, *Appendix 5—table 2*). The 95% sensitivity threshold was equal to 6 SNPs for the two shorter infectious periods (i.e. the higher sampling rates), while it was 4 SNPs for the two longer infectious periods, although this was caused by a greater dispersion of clustering rates in the former two, and there were no differences in the median values. Indeed, the minimum threshold for which 100% of the samples were clustered in at least 95% of the simulations showed no trend (14, 15, 16, 15, and 16 SNPs from the shortest to the longest infectious period). Finally, the mean of the TBL distribution was 0.38 or 0.39 for all settings.

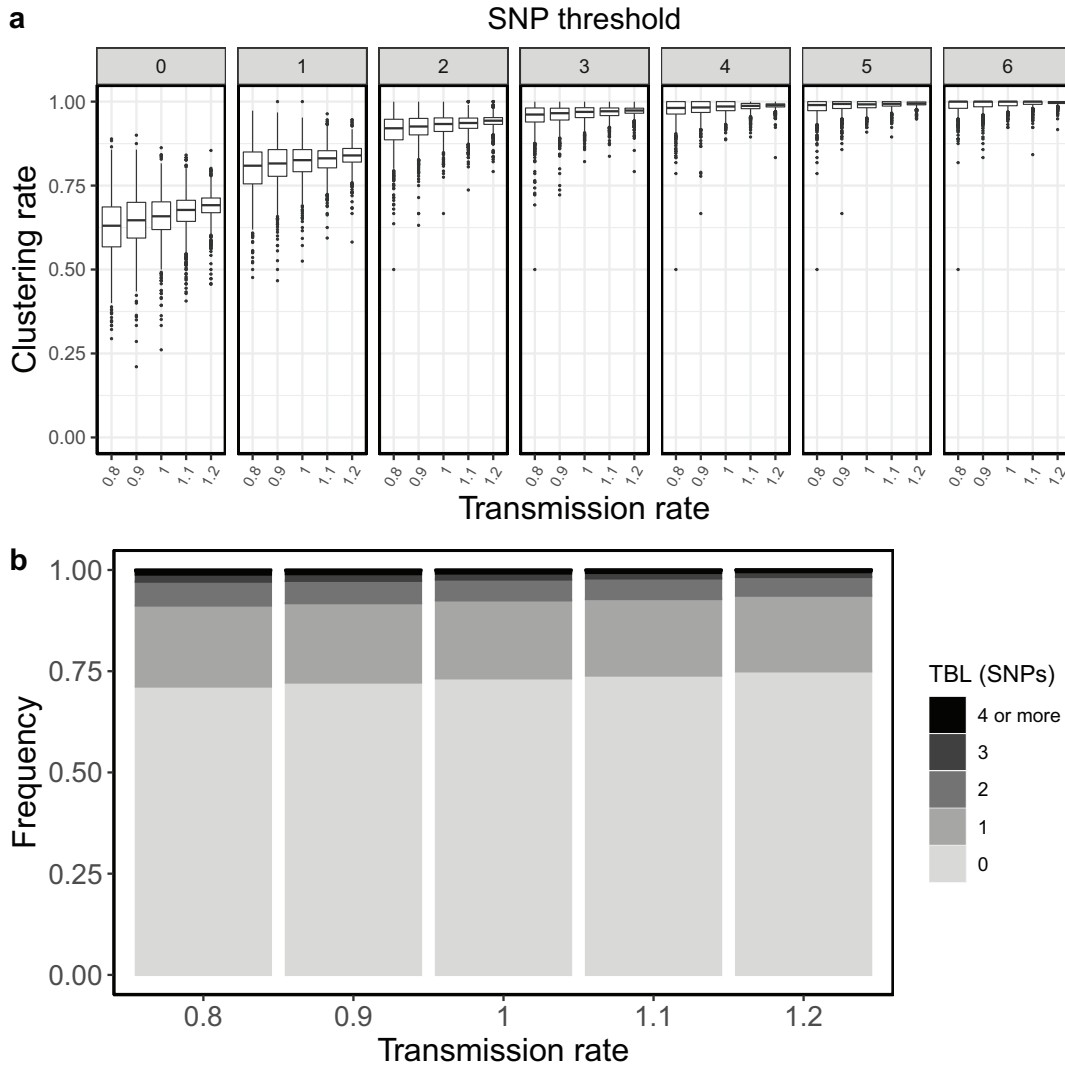

**Appendix 5—figure 1.** Clustering rates and TBL distributions for different transmission rates and a minimum simulated tree size of 100 tips. (**a**) Clustering rates with different SNP thresholds. Only SNP thresholds up to the highest 95% sensitivity threshold are plotted (i.e. for higher thresholds more than 95% of samples are clustered in more than 95% of simulations for all settings). (**b**) Overall TBL distributions computed by merging all simulations.

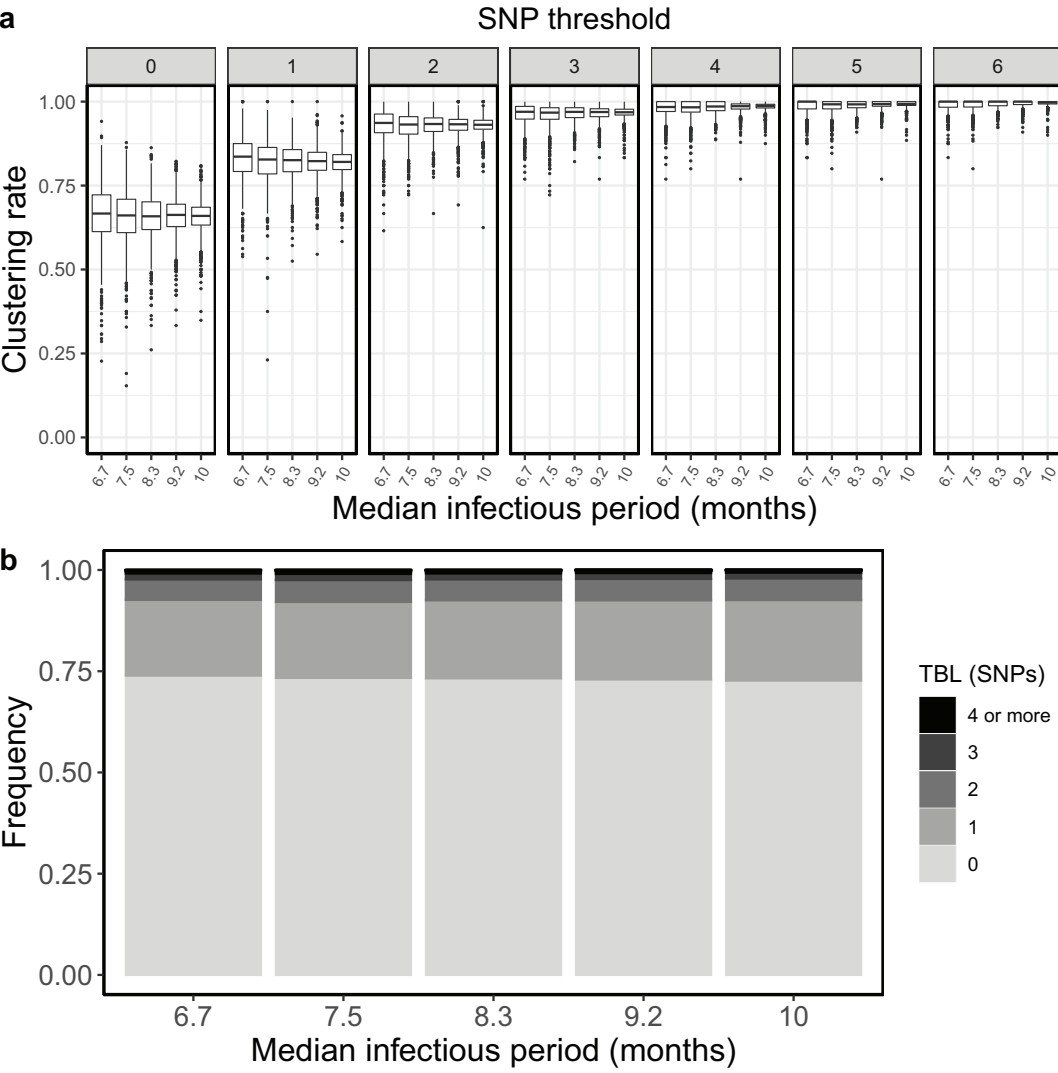

**Appendix 5—figure 2.** Clustering rates and TBL distributions for different sampling rates (and therefore infectious periods), and a minimum simulated tree size of 100 tips. (**a**) Clustering rates with different SNP thresholds. Only SNP thresholds up to the highest 95% sensitivity threshold are plotted (i.e. for higher thresholds more than 95% of samples are clustered in more than 95% of simulations for all settings). (**b**) Overall TBL distributions computed by merging all simulations.

**Appendix 5—table 1.** Parameters and results for the different simulated scenarios in the analysis of the transmission rate with minimum tree size = 100.

$\lambda$ : transmission rate, $\varepsilon$: sampling rate, R0 = $\lambda /(\varepsilon+\sigma)$, $\sigma$: death rate, $\psi$: rate of progression to infectiousness, π: molecular clock rate in expected nucleotide changes per site per year, 95% SNP threshold: the minimum SNP threshold for which at least 95% of samples are clustered in at least 95% of simulations, 100% SNP threshold: the minimum SNP threshold for which 100% of samples are clustered in at least 95% of simulations, 95% CI TBL: the confidence interval for the overall TBL distribution, Mean TBL: average terminal branch length for the overall TBL distribution in SNPs, Mean TBL: average of the overall TBL distribution.

| Scenario | $\lambda$ | $\varepsilon$ | R0 | $\sigma$ | $\psi$ | π | 95% SNP threshold | 100% SNP threshold | 95% CI TBL | Mean TBL |
|---|---|---|---|---|---|---|---|---|---|---|
| Fixed ε R0=0.8 | 0.8 | 1 | 0.8 | 0 | 1 | $8 \times 10^{-8}$ | 6 | 17 | 0–3 | 0.43 |

*Appendix 5—table 1 Continued on next page*

*Appendix 5—table 1 Continued*

| Scenario | $\lambda$ | $\varepsilon$ | R0 | $\sigma$ | $\psi$ | $\pi$ | 95% SNP threshold | 100% SNP threshold | 95% CI TBL | Mean TBL |
|---|---|---|---|---|---|---|---|---|---|---|
| Fixed ε R0=0.9 | 0.9 | 1 | 0.9 | 0 | 1 | $8 \times 10^{-8}$ | 5 | 15 | 0–3 | 0.41 |
| Fixed ε R0=1 | 1 | 1 | 1 | 0 | 1 | $8 \times 10^{-8}$ | 5 | 16 | 0–2 | 0.38 |
| Fixed ε R0=1.1 | 1.1 | 1 | 1.1 | 0 | 1 | $8 \times 10^{-8}$ | 4 | 15 | 0–2 | 0.37 |
| Fixed ε R0=1.2 | 1.2 | 1 | 1.2 | 0 | 1 | $8 \times 10^{-8}$ | 4 | 16 | 0–2 | 0.34 |

**Appendix 5—table 2.** Parameters and results for the different simulated scenarios in the analysis of the sampling rate with minimum tree size = 100.

$\lambda$: transmission rate, $\varepsilon$: sampling rate, R0 = $\lambda /(\varepsilon+\sigma)$, $\sigma$: death rate, $\psi$: rate of progression to infectiousness, π: molecular clock rate in expected nucleotide changes per site per year, 95% SNP threshold: the minimum SNP threshold for which at least 95% of samples are clustered in at least 95% of simulations, 100% SNP threshold: the minimum SNP threshold for which 100% of samples are clustered in at least 95% of simulations, 95% CI TBL: the confidence interval for the overall TBL distribution, Mean TBL: average terminal branch length for the overall TBL distribution in SNPs, Mean TBL: average of the overall TBL distribution.

| Scenario | $\lambda$ | $\varepsilon$ | R0 | $\sigma$ | $\psi$ | $\pi$ | 95% SNP threshold | 100% SNP threshold | 95% CI TBL | Mean TBL |
|---|---|---|---|---|---|---|---|---|---|---|
| Fixed λ R0=0.8 | 1 | 1.25 | 0.8 | 0 | 1 | $8 \times 10^{-8}$ | 6 | 14 | 0–2 | 0.38 |
| Fixed λ R0=0.9 | 1 | 1.11111 | 0.9 | 0 | 1 | $8 \times 10^{-8}$ | 6 | 15 | 0–2 | 0.39 |
| Fixed λ R0=1 | 1 | 1 | 1 | 0 | 1 | $8 \times 10^{-8}$ | 5 | 16 | 0–2 | 0.38 |
| Fixed λ R0=1.1 | 1 | 0.90909 | 1.1 | 0 | 1 | $8 \times 10^{-8}$ | 4 | 15 | 0–2 | 0.38 |
| Fixed λ R0=1.2 | 1 | 0.83333 | 1.2 | 0 | 1 | $8 \times 10^{-8}$ | 4 | 16 | 0–2 | 0.38 |

I investigated whether the threshold on the minimum tree size could bias the analyses of scenarios with different R0. In the original analyses I accepted only simulations that resulted in at least 100 tips over the whole simulated period. I repeated these analyses using 200, 50 and 25 as alternative minimum tips thresholds. For both transmission and sampling rates I confirmed the results of the original analysis for all thresholds. Increasing transmission rates (and therefore R0) resulted in larger clustering rates and shorter terminal branches. This trend was clear for all thresholds, and it was monotone over the entire range of R0 values (0.8–1.2). Conversely, the sampling rate did not strongly affect clustering rates and TBL. There might be a weak trend towards lower clustering rates for lower sampling rates, this was similar for all thresholds, and it was mostly visible for R0 values between 0.8 and 1 (*Appendix 5—figures 1–8*, *Appendix 5—Tables 1–8*).

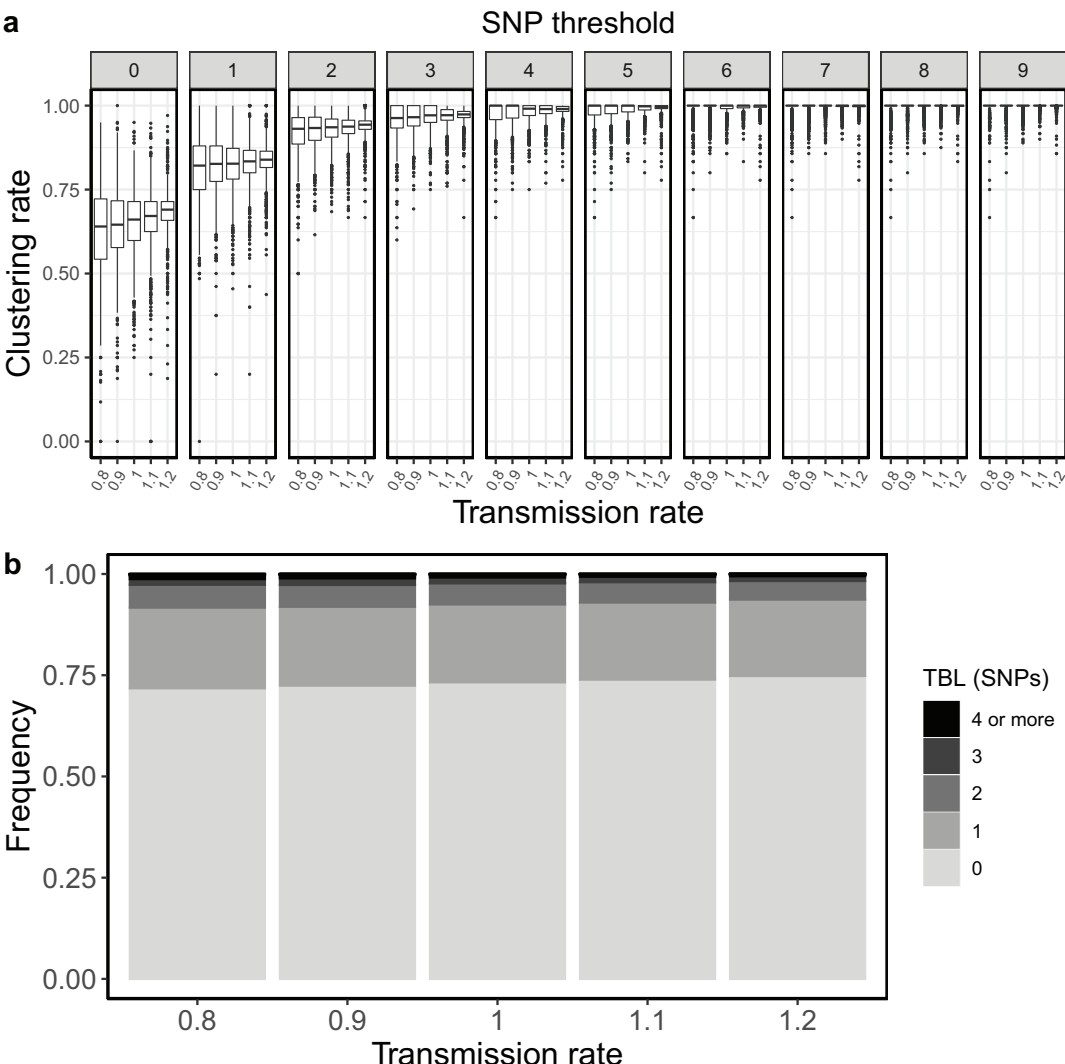

**Appendix 5—figure 3.** Clustering rates and TBL distributions for different transmission rates and a minimum simulated tree size of 25 tips. (**a**) Clustering rates with different SNP thresholds. Only SNP thresholds up to the highest 95% sensitivity threshold are plotted (i.e. for higher thresholds more than 95% of samples are clustered in more than 95% of simulations for all settings). (**b**) Overall TBL distributions computed by merging all simulations.

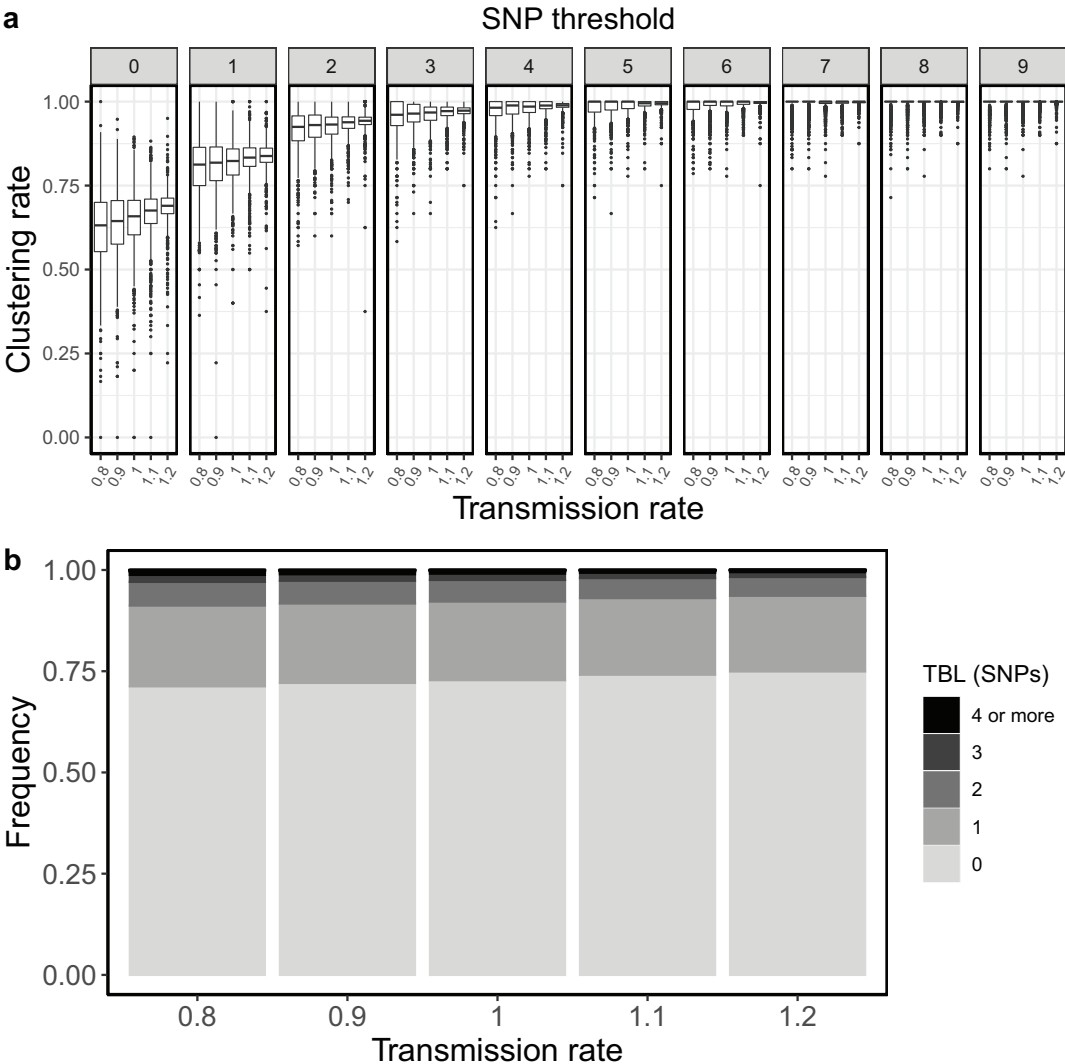

**Appendix 5—figure 4.** Clustering rates and TBL distributions for different transmission rates and a minimum simulated tree size of 50 tips. (**a**) Clustering rates with different SNP thresholds. Only SNP thresholds up to the highest 95% sensitivity threshold are plotted (i.e. for higher thresholds more than 95% of samples are clustered in more than 95% of simulations for all settings). (**b**) Overall TBL distributions computed by merging all simulations.

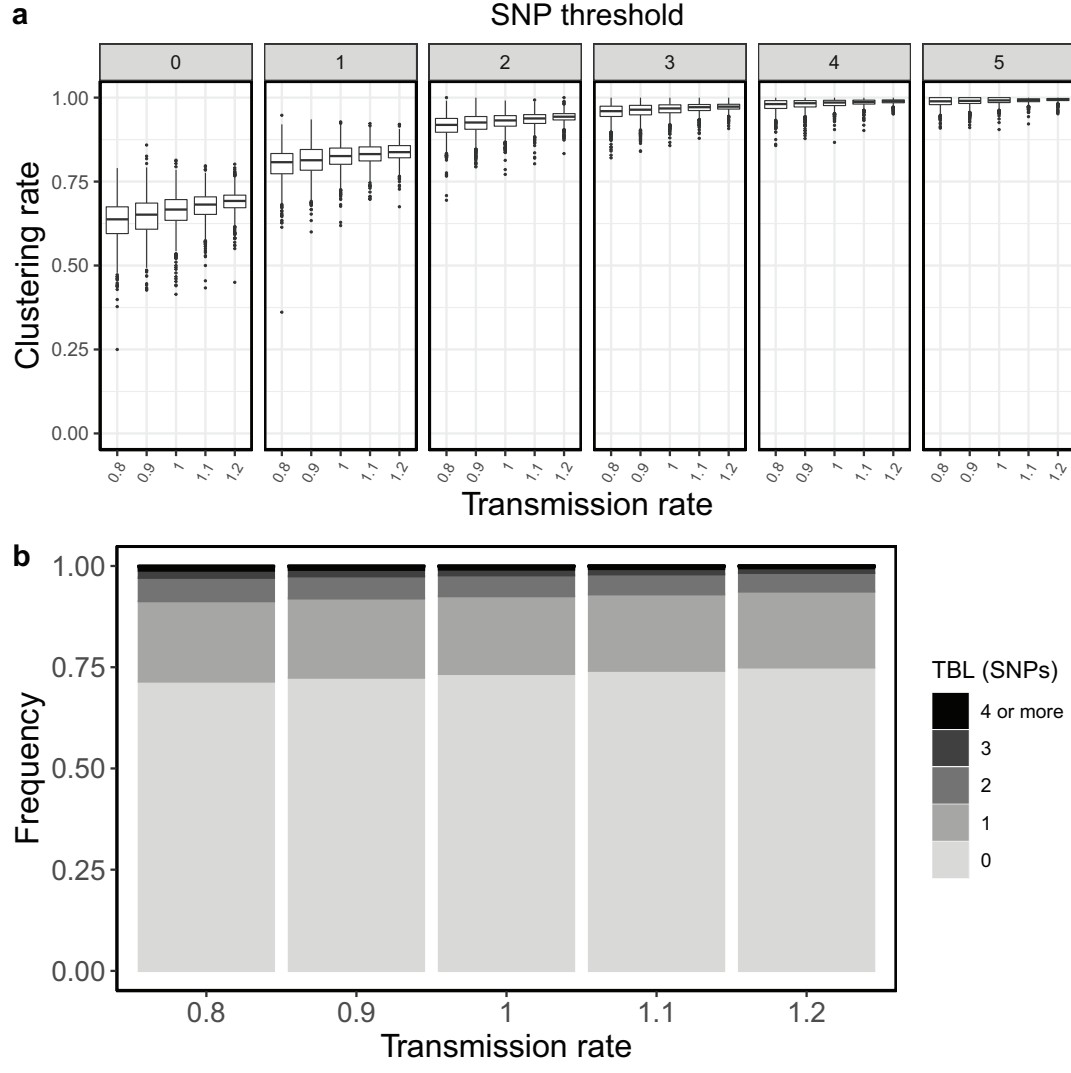

**Appendix 5—figure 5.** Clustering rates and TBL distributions for different transmission rates and a minimum simulated tree size of 200 tips. (**a**) Clustering rates with different SNP thresholds. Only SNP thresholds up to the highest 95% sensitivity threshold are plotted (i.e. for higher thresholds more than 95% of samples are clustered in more than 95% of simulations for all settings). (**b**) Overall TBL distributions computed by merging all simulations.

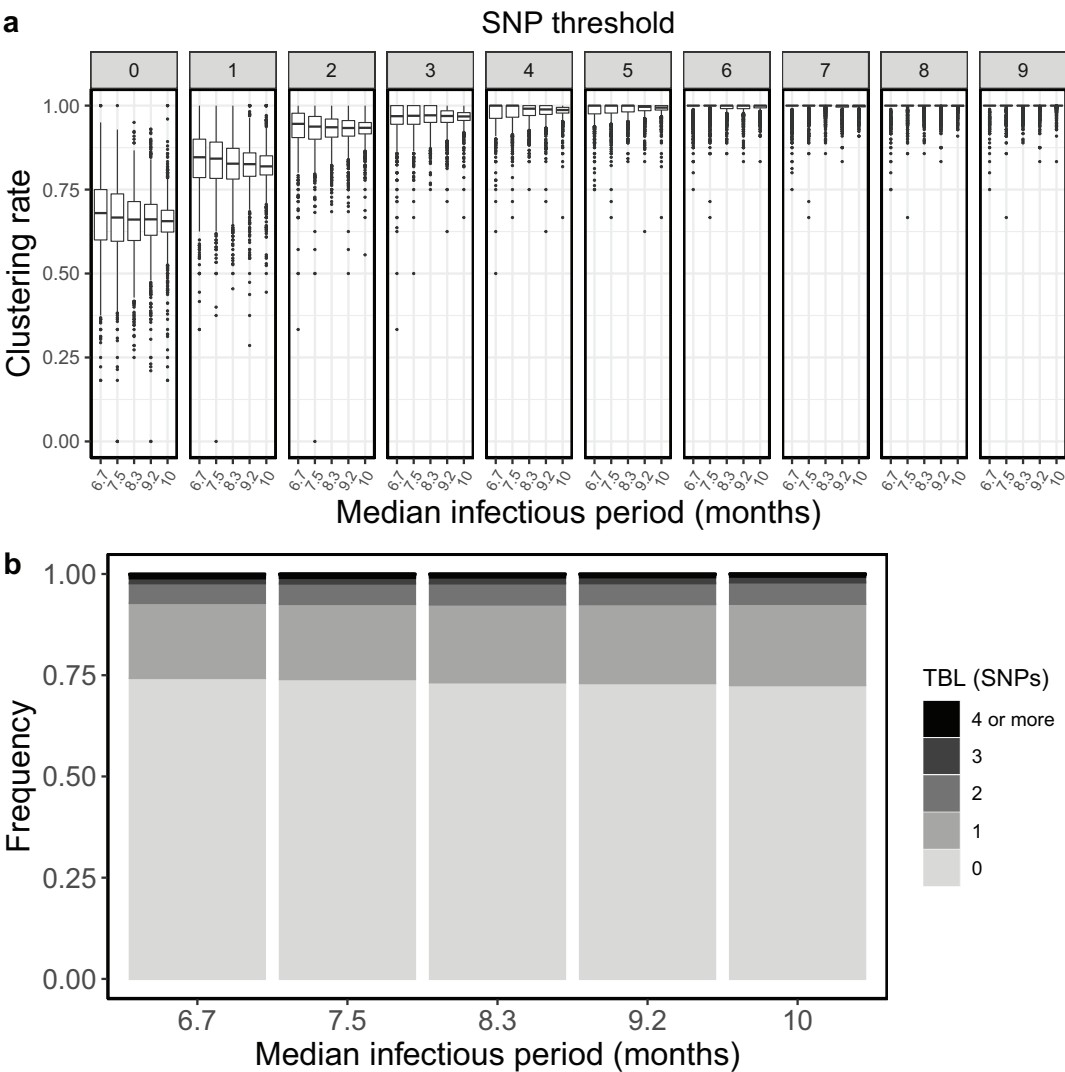

**Appendix 5—figure 6.** Clustering rates and TBL distributions for different sampling rates (and therefore infectious periods), and a minimum simulated tree size of 25 tips. (**a**) Clustering rates with different SNP thresholds. Only SNP thresholds up to the highest 95% sensitivity threshold are plotted (i.e. for higher thresholds more than 95% of samples are clustered in more than 95% of simulations for all settings). (**b**) Overall TBL distributions computed by merging all simulation.

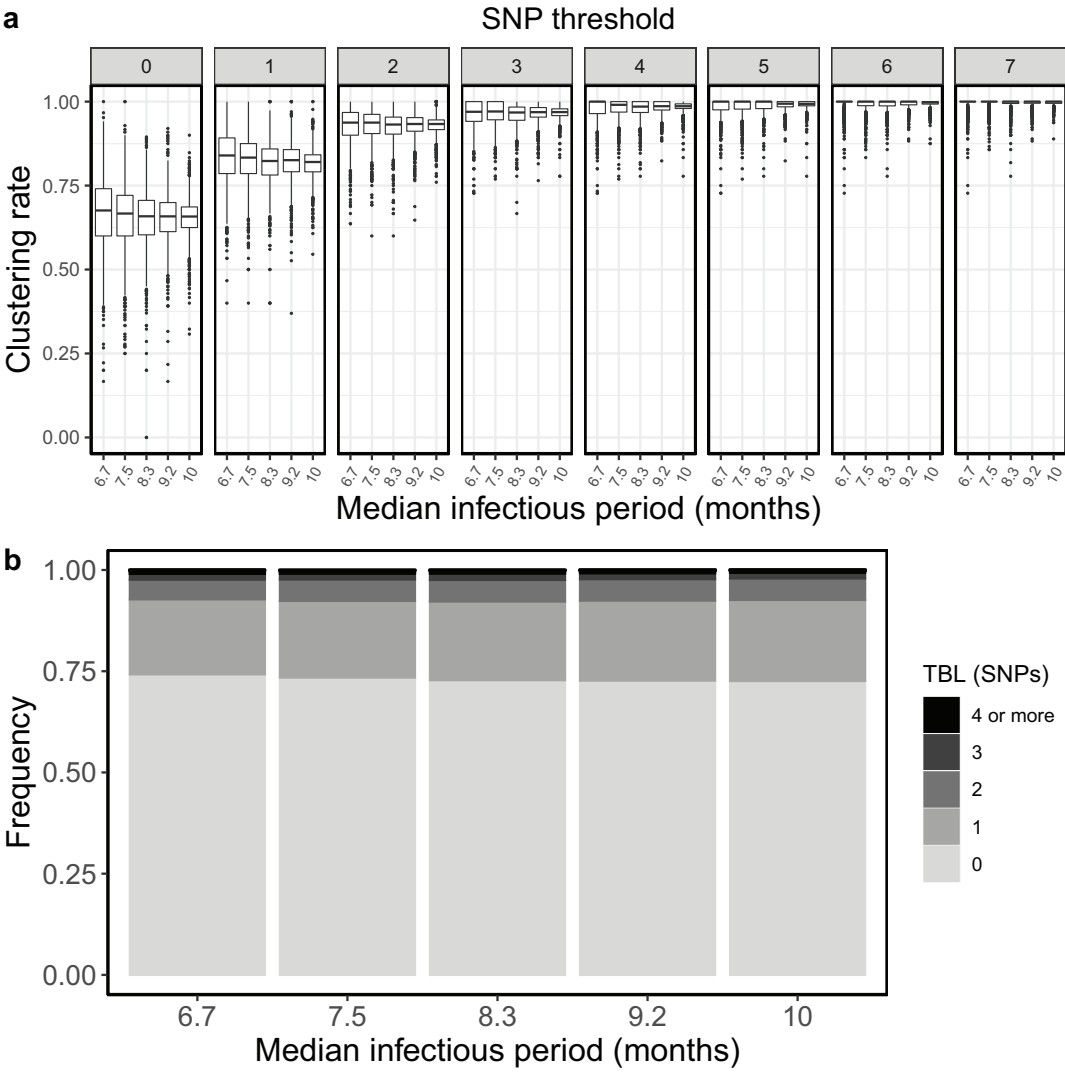

**Appendix 5—figure 7.** Clustering rates and TBL distributions for different sampling rates (and therefore infectious periods), and a minimum simulated tree size of 50 tips. (**a**) Clustering rates with different SNP thresholds. Only SNP thresholds up to the highest 95% sensitivity threshold are plotted (i.e. for higher thresholds more than 95% of samples are clustered in more than 95% of simulations for all settings). (**b**) Overall TBL distributions computed by merging all simulation.

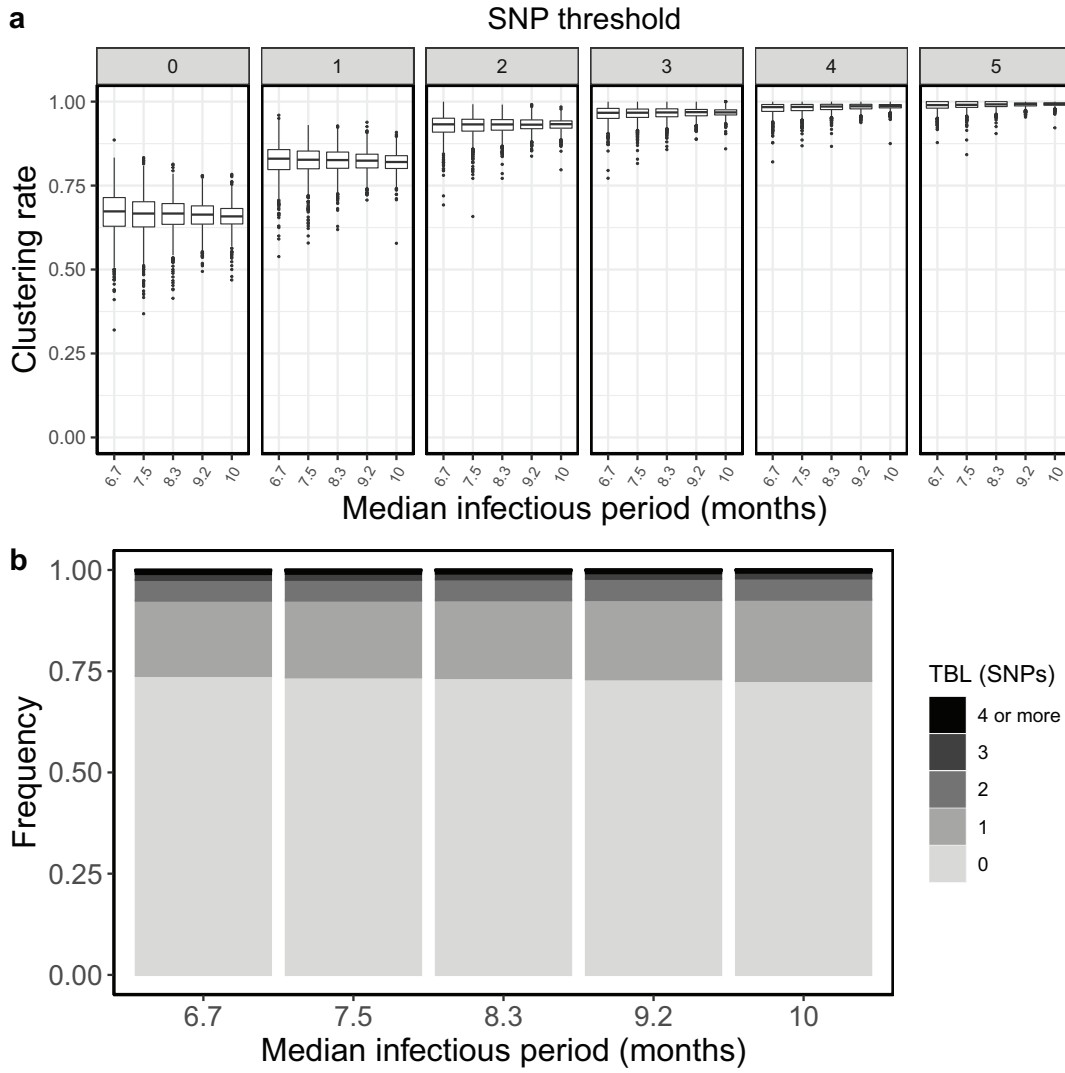

**Appendix 5—figure 8.** Clustering rates and TBL distributions for different sampling rates (and therefore infectious periods), and a minimum simulated tree size of 200 tips. (**a**) Clustering rates with different SNP thresholds. Only SNP thresholds up to the highest 95% sensitivity threshold are plotted (i.e. for higher thresholds more than 95% of samples are clustered in more than 95% of simulations for all settings). (**b**) Overall TBL distributions computed by merging all simulation.

To explore the influence of different thresholds more directly, I compared the results obtained with the same settings and different thresholds. To limit the number of supplementary tables and figures I focused on the scenarios with R0=0.8, 1, and 1.2. When I compared the results obtained for R0=0.8, the scenario with potentially the largest bias, I found that the clustering rates and TBL were not identical when using different thresholds on the minimum tree size. There was a weak trend towards lower clustering rates for larger thresholds (on the minimum tree size), although this was not monotone for all SNP thresholds (*Appendix 5—figures 9–13*, *Appendix 5—Tables 9–13*). Additionally, for one of the analyses with R0=0.8 ($\lambda$ =1 and $\varepsilon$=1.25) I found that larger thresholds also resulted in longer terminal branches, although the difference between the mean of the TBL distributions was smaller than 0.01 SNPs (0.3729, 0.3768, 0.3773, and 0.3815 for a threshold of 25, 50, 100, and 200, respectively). These trends were not present for larger values of R0, indicating that they are indeed a bias due to the minimum tree size, which affect scenarios with shrinking populations.

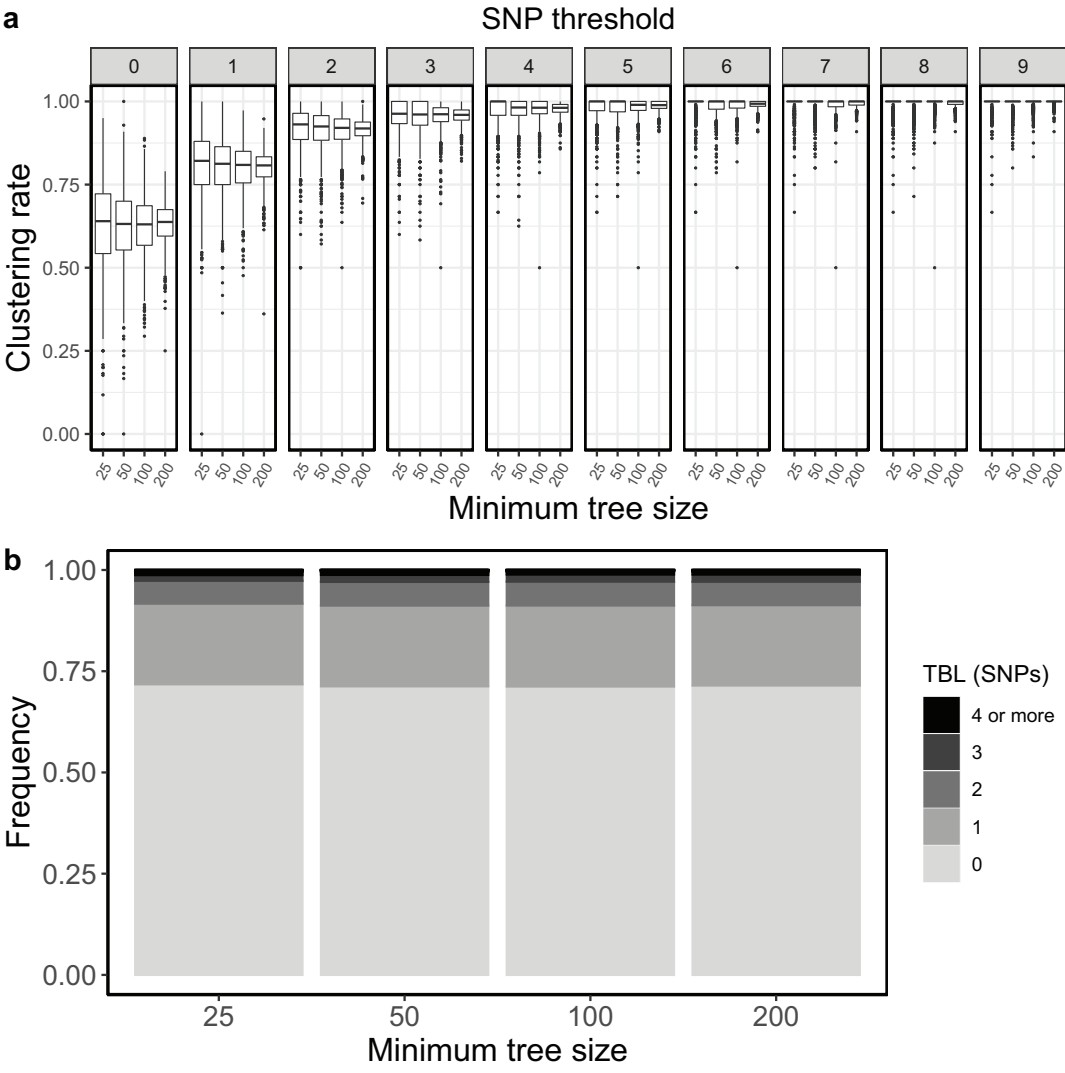

**Appendix 5—figure 9.** Clustering rates and TBL distributions for scenarios with transmission rate = 0.8, and different thresholds on the minimum tree size. (**a**) Clustering rates with different SNP thresholds. Only SNP thresholds up to the highest 95% sensitivity threshold are plotted (i.e. for higher thresholds more than 95% of samples are clustered in more than 95% of simulations for all settings). (**b**) Overall TBL distributions computed by merging all simulations.

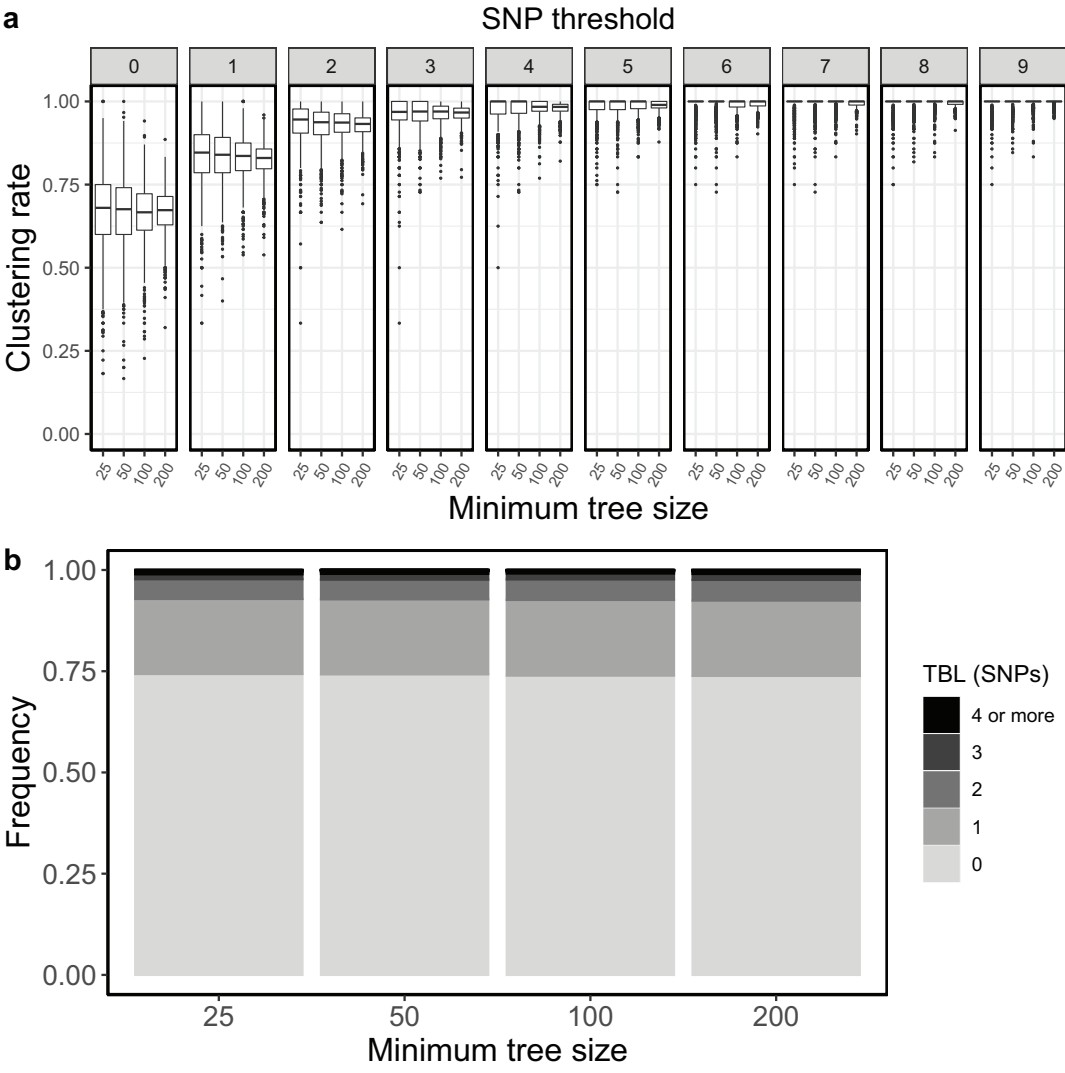

**Appendix 5—figure 10.** Clustering rates and TBL distributions for scenarios with sampling rate = 1.25, and different thresholds on the minimum tree size. (**a**) Clustering rates with different SNP thresholds. Only SNP thresholds up to the highest 95% sensitivity threshold are plotted (i.e. for higher thresholds more than 95% of samples are clustered in more than 95% of simulations for all settings). (**b**) Overall TBL distributions computed by merging all simulations.

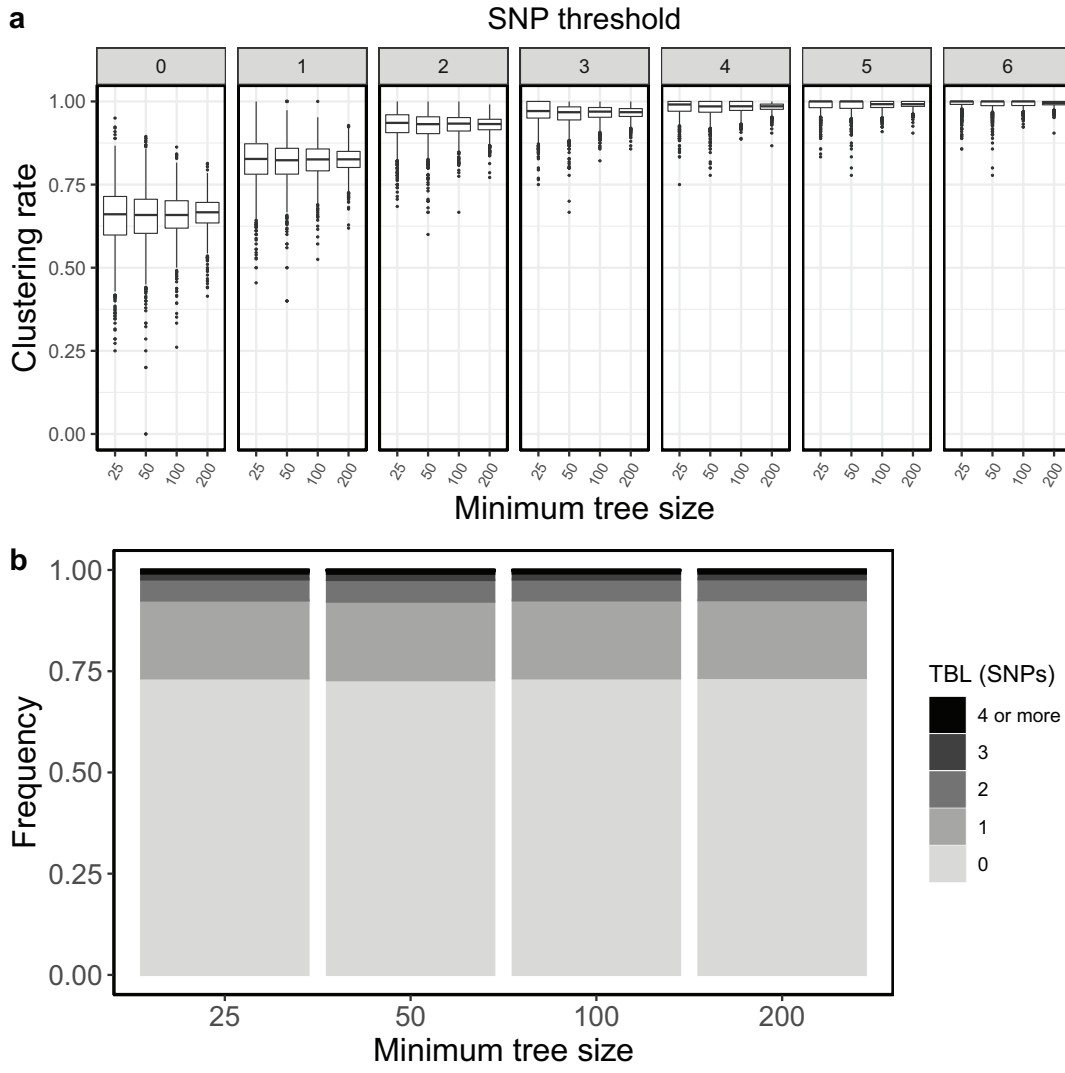

**Appendix 5—figure 11.** Clustering rates and TBL distributions for scenarios with transmission rate = 1, sampling rate = 1, and different thresholds on the minimum tree size. (**a**) Clustering rates with different SNP thresholds. Only SNP thresholds up to the highest 95% sensitivity threshold are plotted (i.e. for higher thresholds more than 95% of samples are clustered in more than 95% of simulations for all settings). (**b**) Overall TBL distributions computed by merging all simulations.

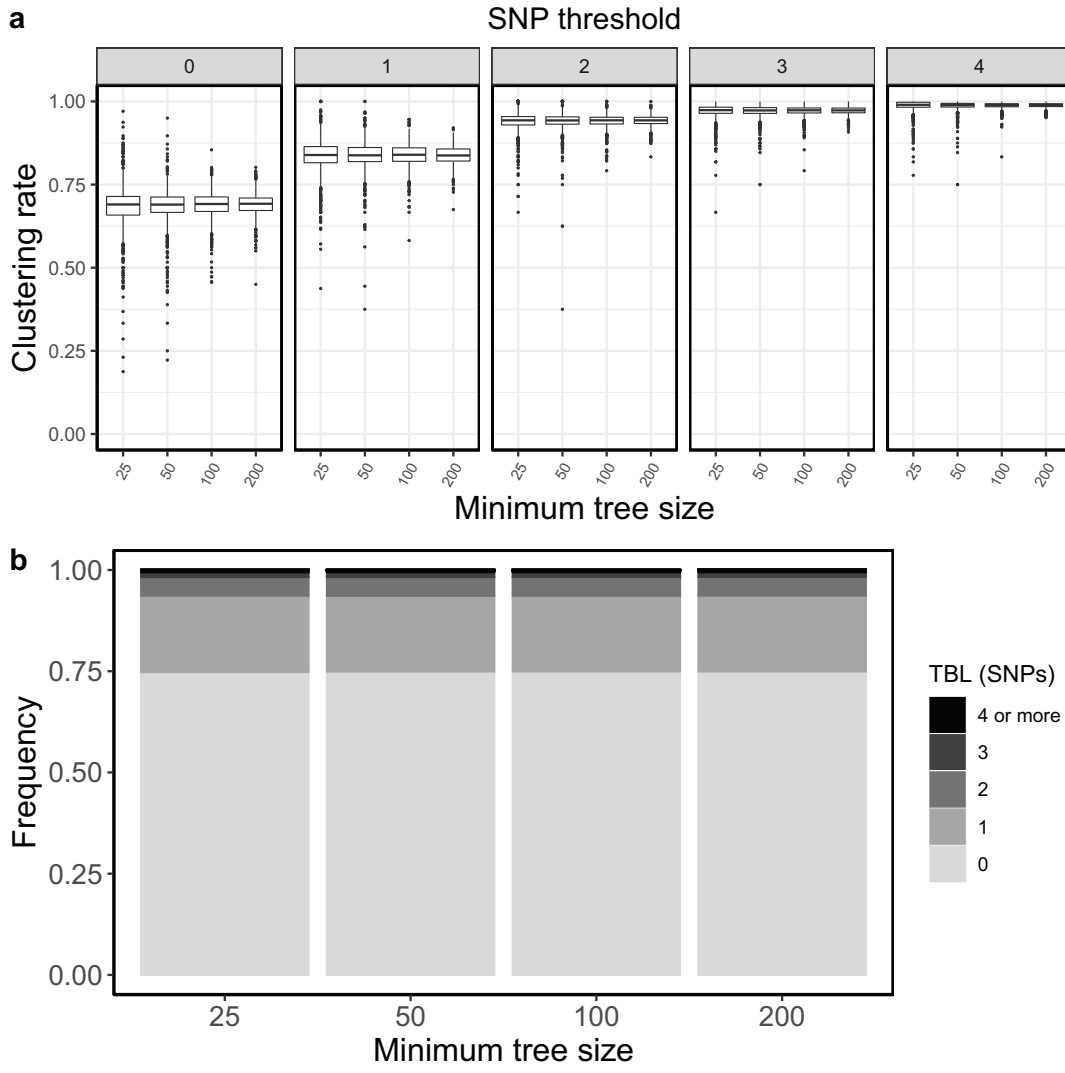

**Appendix 5—figure 12.** Clustering rates and TBL distributions for scenarios with transmission rate = 1.2, and different thresholds on the minimum tree size. (**a**) Clustering rates with different SNP thresholds. Only SNP thresholds up to the highest 95% sensitivity threshold are plotted (i.e. for higher thresholds more than 95% of samples are clustered in more than 95% of simulations for all settings). (**b**) Overall TBL distributions computed by merging all simulations.

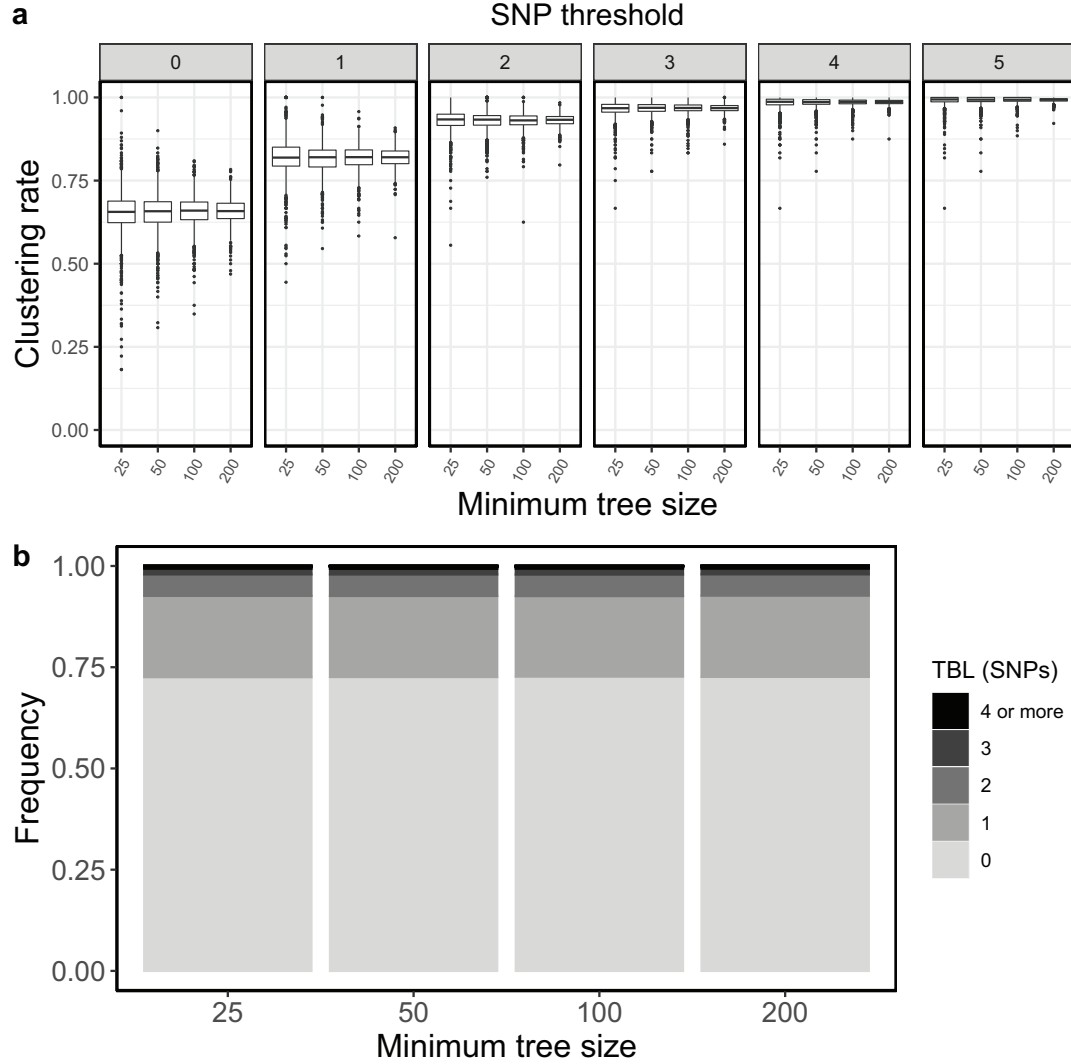

**Appendix 5—figure 13.** Clustering rates and TBL distributions for scenarios with sampling rate = 0.83333, and different thresholds on the minimum tree size. (**a**) Clustering rates with different SNP thresholds. Only SNP thresholds up to the highest 95% sensitivity threshold are plotted (i.e. for higher thresholds more than 95% of samples are clustered in more than 95% of simulations for all settings). (**b**) Overall TBL distributions computed by merging all simulations.

These results suggest that the threshold on the minimum tree size can bias the outcome of some analyses. However, with the settings used in this study this bias was negligible, and the results were robust to different thresholds. For the transmission rate, the trend towards larger clustering rates for higher transmission rates was clear also for scenarios with R0 equal or greater than one, these scenarios are scarcely affected by the bias, further indicating that this is a genuine trend due to the epidemiological process (*Appendix 5—figures 1 and 3–5*). Conversely, for the sampling rate, the observed trend was evident mostly for scenarios with R0 smaller than one, suggesting that it was caused by the threshold on the tree size (*Appendix 5—figures 2 and 6–8*). In any case, the differences were rather small, and I conclude that if there is any effect of the sampling rate on clustering and TBL, this is extremely weak.

**Appendix 5—table 3.** Parameters and results for the different simulated scenarios in the analysis of the transmission rate with minimum tree size = 25.
$\lambda$: transmission rate, $\varepsilon$: sampling rate, R0 = $\lambda /(\varepsilon+\sigma)$, $\sigma$: death rate, $\psi$: rate of progression to infectiousness, $\pi$: molecular clock rate in expected nucleotide changes per site per year, 95% SNP threshold: the minimum SNP threshold for which at least 95% of samples are clustered in at least

95% of simulations, 100% SNP threshold: the minimum SNP threshold for which 100% of samples are clustered in at least 95% of simulations, 95% CI TBL: the confidence interval for the overall TBL distribution, Mean TBL: average terminal branch length for the overall TBL distribution in SNPs, Mean TBL: average of the overall TBL distribution.

| Scenario | $\lambda$ | $\varepsilon$ | R0 | $\sigma$ | $\psi$ | $\pi$ | 95% SNP threshold | 100% SNP threshold | 95% CI TBL | Mean TBL |
|---|---|---|---|---|---|---|---|---|---|---|
| Fixed ε R0=0.8 | 0.8 | 1 | 0.8 | 0 | 1 | $8 \times 10^{-8}$ | 9 | 12 | 0–3 | 0.42 |
| Fixed ε R0=0.9 | 0.9 | 1 | 0.9 | 0 | 1 | $8 \times 10^{-8}$ | 7 | 13 | 0–3 | 0.41 |
| Fixed ε R0=1 | 1 | 1 | 1 | 0 | 1 | $8 \times 10^{-8}$ | 6 | 13 | 0–2 | 0.39 |
| Fixed ε R0=1.1 | 1.1 | 1 | 1.1 | 0 | 1 | $8 \times 10^{-8}$ | 5 | 13 | 0–2 | 0.37 |
| Fixed ε R0=1.2 | 1.2 | 1 | 1.2 | 0 | 1 | $8 \times 10^{-8}$ | 4 | 14 | 0–2 | 0.34 |

**Appendix 5—table 4.** Parameters and results for the different simulated scenarios in the analysis of the transmission rate with minimum tree size = 50.

$\lambda$ : transmission rate, $\varepsilon$: sampling rate, R0 = $\lambda /(\varepsilon+\sigma)$, $\sigma$: death rate, $\psi$: rate of progression to infectiousness, π: molecular clock rate in expected nucleotide changes per site per year, 95% SNP threshold: the minimum SNP threshold for which at least 95% of samples are clustered in at least 95% of simulations, 100% SNP threshold: the minimum SNP threshold for which 100% of samples are clustered in at least 95% of simulations, 95% CI TBL: the confidence interval for the overall TBL distribution, Mean TBL: average terminal branch length for the overall TBL distribution in SNPs, Mean TBL: average of the overall TBL distribution.

| Scenario | $\lambda$ | $\varepsilon$ | R0 | $\sigma$ | $\psi$ | $\pi$ | 95% SNP threshold | 100% SNP threshold | 95% CI TBL | Mean TBL |
|---|---|---|---|---|---|---|---|---|---|---|
| Fixed ε R0=0.8 | 0.8 | 1 | 0.8 | 0 | 1 | $8 \times 10^{-8}$ | 9 | 15 | 0–3 | 0.43 |
| Fixed ε R0=0.9 | 0.9 | 1 | 0.9 | 0 | 1 | $8 \times 10^{-8}$ | 8 | 14 | 0–3 | 0.42 |
| Fixed ε R0=1 | 1 | 1 | 1 | 0 | 1 | $8 \times 10^{-8}$ | 6 | 15 | 0–2 | 0.40 |
| Fixed ε R0=1.1 | 1.1 | 1 | 1.1 | 0 | 1 | $8 \times 10^{-8}$ | 4 | 14 | 0–2 | 0.36 |
| Fixed ε R0=1.2 | 1.2 | 1 | 1.2 | 0 | 1 | $8 \times 10^{-8}$ | 4 | 15 | 0–2 | 0.34 |

**Appendix 5—table 5.** Parameters and results for the different simulated scenarios in the analysis of the transmission rate with minimum tree size = 200.

$\lambda$ : transmission rate, $\varepsilon$: sampling rate, R0 = $\lambda /(\varepsilon+\sigma)$, $\sigma$: death rate, $\psi$: rate of progression to infectiousness, π: molecular clock rate in expected nucleotide changes per site per year, 95% SNP threshold: the minimum SNP threshold for which at least 95% of samples are clustered in at least 95% of simulations, 100% SNP threshold: the minimum SNP threshold for which 100% of samples are clustered in at least 95% of simulations, 95% CI TBL: the confidence interval for the overall TBL distribution, Mean TBL: average terminal branch length for the overall TBL distribution in SNPs, Mean TBL: average of the overall TBL distribution.

| Scenario | $\lambda$ | $\varepsilon$ | R0 | $\sigma$ | $\psi$ | $\pi$ | 95% SNP threshold | 100% SNP threshold | 95% CI TBL | Mean TBL |
|---|---|---|---|---|---|---|---|---|---|---|
| Fixed ε R0=0.8 | 0.8 | 1 | 0.8 | 0 | 1 | $8 \times 10^{-8}$ | 5 | 16 | 0–3 | 0.42 |

*Appendix 5—table 5 Continued*

| Scenario | $\lambda$ | $\varepsilon$ | R0 | $\sigma$ | $\psi$ | $\pi$ | 95% SNP threshold | 100% SNP threshold | 95% CI TBL | Mean TBL |
|---|---|---|---|---|---|---|---|---|---|---|
| Fixed ε R0=0.9 | 0.9 | 1 | 0.9 | 0 | 1 | $8 \times 10^{-8}$ | 5 | 17 | 0–3 | 0.40 |
| Fixed ε R0=1 | 1 | 1 | 1 | 0 | 1 | $8 \times 10^{-8}$ | 4 | 15 | 0–2 | 0.38 |
| Fixed ε R0=1.1 | 1.1 | 1 | 1.1 | 0 | 1 | $8 \times 10^{-8}$ | 4 | 17 | 0–2 | 0.37 |
| Fixed ε R0=1.2 | 1.2 | 1 | 1.2 | 0 | 1 | $8 \times 10^{-8}$ | 3 | 17 | 0–2 | 0.34 |

**Appendix 5—table 6.** Parameters and results for the different simulated scenarios in the analysis of the sampling rate with minimum tree size = 25.

$\lambda$ : transmission rate, $\varepsilon$: sampling rate, R0 = $\lambda /(\varepsilon+\sigma)$, $\sigma$: death rate, $\psi$: rate of progression to infectiousness, π: molecular clock rate in expected nucleotide changes per site per year, 95% SNP threshold: the minimum SNP threshold for which at least 95% of samples are clustered in at least 95% of simulations, 100% SNP threshold: the minimum SNP threshold for which 100% of samples are clustered in at least 95% of simulations, 95% CI TBL: the confidence interval for the overall TBL distribution, Mean TBL: average terminal branch length for the overall TBL distribution in SNPs, Mean TBL: average of the overall TBL distribution.

| Scenario | $\lambda$ | $\varepsilon$ | R0 | $\sigma$ | $\psi$ | $\pi$ | 95% SNP threshold | 100% SNP threshold | 95% CI TBL | Mean TBL |
|---|---|---|---|---|---|---|---|---|---|---|
| Fixed $\lambda$ R0=0.8 | 1 | 1.25 | 0.8 | 0 | 1 | $8 \times 10^{-8}$ | 9 | 12 | 0–2 | 0.37 |
| Fixed $\lambda$ R0=0.9 | 1 | 1.11111 | 0.9 | 0 | 1 | $8 \times 10^{-8}$ | 8 | 12 | 0–2 | 0.38 |
| Fixed $\lambda$ R0=1 | 1 | 1 | 1 | 0 | 1 | $8 \times 10^{-8}$ | 6 | 13 | 0–2 | 0.39 |
| Fixed $\lambda$ R0=1.1 | 1 | 0.90909 | 1.1 | 0 | 1 | $8 \times 10^{-8}$ | 6 | 15 | 0–2 | 0.39 |
| Fixed $\lambda$ R0=1.2 | 1 | 0.83333 | 1.2 | 0 | 1 | $8 \times 10^{-8}$ | 5 | 15 | 0–2 | 0.38 |

**Appendix 5—table 7.** Parameters and results for the different simulated scenarios in the analysis of the sampling rate with minimum tree size = 50.

$\lambda$ : transmission rate, $\varepsilon$: sampling rate, R0 = $\lambda /(\varepsilon+\sigma)$, $\sigma$: death rate, $\psi$: rate of progression to infectiousness, π: molecular clock rate in expected nucleotide changes per site per year, 95% SNP threshold: the minimum SNP threshold for which at least 95% of samples are clustered in at least 95% of simulations, 100% SNP threshold: the minimum SNP threshold for which 100% of samples are clustered in at least 95% of simulations, 95% CI TBL: the confidence interval for the overall TBL distribution, Mean TBL: average terminal branch length for the overall TBL distribution in SNPs, Mean TBL: average of the overall TBL distribution.

| Scenario | $\lambda$ | $\varepsilon$ | R0 | $\sigma$ | $\psi$ | $\pi$ | 95% SNP threshold | 100% SNP threshold | 95% CI TBL | Mean TBL |
|---|---|---|---|---|---|---|---|---|---|---|
| Fixed $\lambda$ R0=0.8 | 1 | 1.25 | 0.8 | 0 | 1 | $8 \times 10^{-8}$ | 7 | 13 | 0–2 | 0.38 |
| Fixed $\lambda$ R0=0.9 | 1 | 1.11111 | 0.9 | 0 | 1 | $8 \times 10^{-8}$ | 7 | 14 | 0–2 | 0.39 |
| Fixed $\lambda$ R0=1 | 1 | 1 | 1 | 0 | 1 | $8 \times 10^{-8}$ | 6 | 15 | 0–2 | 0.40 |
| Fixed $\lambda$ R0=1.1 | 1 | 0.90909 | 1.1 | 0 | 1 | $8 \times 10^{-8}$ | 5 | 15 | 0–2 | 0.39 |

Appendix 5—table 7 Continued

| Scenario | $\lambda$ | $\varepsilon$ | R0 | $\sigma$ | $\psi$ | $\pi$ | 95% SNP threshold | 100% SNP threshold | 95% CI TBL | Mean TBL |
|---|---|---|---|---|---|---|---|---|---|---|
| Fixed $\lambda$ R0=1.2 | 1 | 0.83333 | 1.2 | 0 | 1 | $8 \times 10^{-8}$ | 4 | 16 | 0–2 | 0.38 |

**Appendix 5—table 8.** Parameters and results for the different simulated scenarios in the analysis of the sampling rate with minimum tree size = 200.

$\lambda$: transmission rate, $\varepsilon$: sampling rate, R0 = $\lambda /(\varepsilon+\sigma)$, $\sigma$: death rate, $\psi$: rate of progression to infectiousness, $\pi$: molecular clock rate in expected nucleotide changes per site per year, 95% SNP threshold: the minimum SNP threshold for which at least 95% of samples are clustered in at least 95% of simulations, 100% SNP threshold: the minimum SNP threshold for which 100% of samples are clustered in at least 95% of simulations, 95% CI TBL: the confidence interval for the overall TBL distribution, Mean TBL: average terminal branch length for the overall TBL distribution in SNPs, Mean TBL: average of the overall TBL distribution.

| Scenario | $\lambda$ | $\varepsilon$ | R0 | $\sigma$ | $\psi$ | $\pi$ | 95% SNP threshold | 100% SNP threshold | 95% CI TBL | Mean TBL |
|---|---|---|---|---|---|---|---|---|---|---|
| Fixed $\lambda$ R0=0.8 | 1 | 1.25 | 0.8 | 0 | 1 | $8 \times 10^{-8}$ | 5 | 16 | 0–2 | 0.38 |
| Fixed $\lambda$ R0=0.9 | 1 | 1.11111 | 0.9 | 0 | 1 | $8 \times 10^{-8}$ | 5 | 17 | 0–2 | 0.38 |
| Fixed $\lambda$ R0=1 | 1 | 1 | 1 | 0 | 1 | $8 \times 10^{-8}$ | 4 | 15 | 0–2 | 0.38 |
| Fixed $\lambda$ R0=1.1 | 1 | 0.90909 | 1.1 | 0 | 1 | $8 \times 10^{-8}$ | 4 | 15 | 0–2 | 0.38 |
| Fixed $\lambda$ R0=1.2 | 1 | 0.83333 | 1.2 | 0 | 1 | $8 \times 10^{-8}$ | 4 | 16 | 0–2 | 0.38 |

**Appendix 5—table 9.** Parameters and results for scenarios with transmission rate = 0.8 and different thresholds on the minimum tree size.

mts: minimum tree size, $\lambda$: transmission rate, $\varepsilon$: sampling rate, R0 = $\lambda /(\varepsilon+\sigma)$, $\sigma$: death rate, $\psi$: rate of progression to infectiousness, $\pi$: molecular clock rate in expected nucleotide changes per site per year, 95% SNP threshold: the minimum SNP threshold for which at least 95% of samples are clustered in at least 95% of simulations, 100% SNP threshold: the minimum SNP threshold for which 100% of samples are clustered in at least 95% of simulations, 95% CI TBL: the confidence interval for the overall TBL distribution, Mean TBL: average terminal branch length for the overall TBL distribution in SNPs, Mean TBL: average of the overall TBL distribution.

| Scenario | $\lambda$ | $\varepsilon$ | R0 | $\sigma$ | $\psi$ | $\pi$ | 95% SNP threshold | 100% SNP threshold | 95% CI TBL | Mean TBL |
|---|---|---|---|---|---|---|---|---|---|---|
| mts = 25 | 0.8 | 1 | 0.8 | 0 | 1 | $8 \times 10^{-8}$ | 9 | 12 | 0–3 | 0.42 |
| mts = 50 | 0.8 | 1 | 0.8 | 0 | 1 | $8 \times 10^{-8}$ | 9 | 15 | 0–3 | 0.43 |
| mts = 100 | 0.8 | 1 | 0.8 | 0 | 1 | $8 \times 10^{-8}$ | 6 | 17 | 0–3 | 0.43 |
| mts = 200 | 0.8 | 1 | 0.8 | 0 | 1 | $8 \times 10^{-8}$ | 5 | 16 | 0–3 | 0.42 |

**Appendix 5—table 10.** Parameters and results for scenarios with sampling rate = 1.25 and different thresholds on the minimum tree size.

mts: minimum tree size, $\lambda$: transmission rate, $\varepsilon$: sampling rate, R0 = $\lambda /(\varepsilon+\sigma)$, $\sigma$: death rate, $\psi$: rate of progression to infectiousness, $\pi$: molecular clock rate in expected nucleotide changes per site per year, 95% SNP threshold: the minimum SNP threshold for which at least 95% of samples are clustered in at least 95% of simulations, 100% SNP threshold: the minimum SNP threshold for which 100% of samples are clustered in at least 95% of simulations, 95% CI TBL: the confidence interval for the overall TBL distribution, Mean TBL: average terminal branch length for the overall TBL distribution in SNPs, Mean TBL: average of the overall TBL distribution.

| Scenario | $\lambda$ | $\varepsilon$ | R0 | $\sigma$ | $\psi$ | $\pi$ | 95% SNP threshold | 100% SNP threshold | 95% CI TBL | Mean TBL |
|---|---|---|---|---|---|---|---|---|---|---|
| mts = 25 | 1 | 1.25 | 0.8 | 0 | 1 | $8 \times 10^{-8}$ | 9 | 12 | 0–2 | 0.37 |
| mts = 50 | 1 | 1.25 | 0.8 | 0 | 1 | $8 \times 10^{-8}$ | 7 | 13 | 0–2 | 0.38 |
| mts = 100 | 1 | 1.25 | 0.8 | 0 | 1 | $8 \times 10^{-8}$ | 6 | 14 | 0–2 | 0.38 |
| mts = 200 | 1 | 1.25 | 0.8 | 0 | 1 | $8 \times 10^{-8}$ | 5 | 16 | 0–2 | 0.38 |

**Appendix 5—table 11.** Parameters and results for scenarios with transmission rate = 1, sampling rate = 1, and different thresholds on the minimum tree size.

mts: minimum tree size, $\lambda$: transmission rate, $\varepsilon$: sampling rate, R0 = $\lambda/(\varepsilon+\sigma)$, $\sigma$: death rate, $\psi$: rate of progression to infectiousness, π: molecular clock rate in expected nucleotide changes per site per year, 95% SNP threshold: the minimum SNP threshold for which at least 95% of samples are clustered in at least 95% of simulations, 100% SNP threshold: the minimum SNP threshold for which 100% of samples are clustered in at least 95% of simulations, 95% CI TBL: the confidence interval for the overall TBL distribution, Mean TBL: average terminal branch length for the overall TBL distribution in SNPs, Mean TBL: average of the overall TBL distribution.

| Scenario | $\lambda$ | $\varepsilon$ | R0 | $\sigma$ | $\psi$ | $\pi$ | 95% SNP threshold | 100% SNP threshold | 95% CI TBL | Mean TBL |
|---|---|---|---|---|---|---|---|---|---|---|
| mts = 25 | 1 | 1 | 1 | 0 | 1 | $8 \times 10^{-8}$ | 6 | 13 | 0–2 | 0.39 |
| mts = 50 | 1 | 1 | 1 | 0 | 1 | $8 \times 10^{-8}$ | 6 | 15 | 0–2 | 0.40 |
| mts = 100 | 1 | 1 | 1 | 0 | 1 | $8 \times 10^{-8}$ | 5 | 16 | 0–2 | 0.38 |
| mts = 200 | 1 | 1 | 1 | 0 | 1 | $8 \times 10^{-8}$ | 4 | 15 | 0–2 | 0.38 |

**Appendix 5—table 12.** Parameters and results for scenarios with transmission rate = 1.2 and different thresholds on the minimum tree size.

mts: minimum tree size, $\lambda$: transmission rate, $\varepsilon$: sampling rate, R0 = $\lambda/(\varepsilon+\sigma)$, $\sigma$: death rate, $\psi$: rate of progression to infectiousness, π: molecular clock rate in expected nucleotide changes per site per year, 95% SNP threshold: the minimum SNP threshold for which at least 95% of samples are clustered in at least 95% of simulations, 100% SNP threshold: the minimum SNP threshold for which 100% of samples are clustered in at least 95% of simulations, 95% CI TBL: the confidence interval for the overall TBL distribution, Mean TBL: average terminal branch length for the overall TBL distribution in SNPs, Mean TBL: average of the overall TBL distribution.

| Scenario | $\lambda$ | $\varepsilon$ | R0 | $\sigma$ | $\psi$ | $\pi$ | 95% SNP threshold | 100% SNP threshold | 95% CI TBL | Mean TBL |
|---|---|---|---|---|---|---|---|---|---|---|
| mts = 25 | 1.2 | 1 | 1.2 | 0 | 1 | $8 \times 10^{-8}$ | 4 | 14 | 0–2 | 0.34 |
| mts = 50 | 1.2 | 1 | 1.2 | 0 | 1 | $8 \times 10^{-8}$ | 4 | 15 | 0–2 | 0.34 |
| mts = 100 | 1.2 | 1 | 1.2 | 0 | 1 | $8 \times 10^{-8}$ | 4 | 16 | 0–2 | 0.34 |
| mts = 200 | 1.2 | 1 | 1.2 | 0 | 1 | $8 \times 10^{-8}$ | 3 | 17 | 0–2 | 0.34 |

**Appendix 5—table 13.** Parameters and results for scenarios with sampling rate = 0.83333 and different thresholds on the minimum tree size.

mts: minimum tree size, $\lambda$: transmission rate, $\varepsilon$: sampling rate, R0 = $\lambda/(\varepsilon+\sigma)$, $\sigma$: death rate, $\psi$: rate of progression to infectiousness, π: molecular clock rate in expected nucleotide changes per site per year, 95% SNP threshold: the minimum SNP threshold for which at least 95% of samples are clustered in at least 95% of simulations, 100% SNP threshold: the minimum SNP threshold for which 100% of samples are clustered in at least 95% of simulations, 95% CI TBL: the confidence interval for the overall TBL distribution, Mean TBL: average terminal branch length for the overall TBL distribution in SNPs, Mean TBL: average of the overall TBL distribution.

| Scenario | $\lambda$ | $\varepsilon$ | R0 | $\sigma$ | $\psi$ | $\pi$ | 95% SNP threshold | 100% SNP threshold | 95% CI TBL | Mean TBL |
|---|---|---|---|---|---|---|---|---|---|---|
| mts = 25 | 1 | 0.83333 | 1.2 | 0 | 1 | $8 \times 10^{-8}$ | 5 | 15 | 0–2 | 0.38 |
| mts = 50 | 1 | 0.83333 | 1.2 | 0 | 1 | $8 \times 10^{-8}$ | 4 | 16 | 0–2 | 0.38 |
| mts = 100 | 1 | 0.83333 | 1.2 | 0 | 1 | $8 \times 10^{-8}$ | 4 | 16 | 0–2 | 0.38 |
| mts = 200 | 1 | 0.83333 | 1.2 | 0 | 1 | $8 \times 10^{-8}$ | 4 | 16 | 0–2 | 0.38 |

