## [Editor Report]

Grouping pathogen genomes into clusters is a key tool in genomic epidemiology. In this paper, the author takes a simulation-based approach to investigate the epidemiological processes that influence clustering in tuberculosis genomic epidemiology. The simulations explore whether differences in transmission can be detected with clustering-based analysis. This work finds that clustering can be impacted by sampling strategy as well as by changes in transmission and population dynamics, and draws out some interpretations of these results for users of clustering in this field.

---

## [Decision Letter]

**Decision letter after peer review:**

Thank you for submitting your article "Understanding drivers of phylogenetic clustering and terminal branch lengths distribution in epidemics of *Mycobacterium tuberculosis*" for consideration by *eLife*. Your article has been reviewed by 2 peer reviewers, and the evaluation has been overseen by a Reviewing Editor and a Senior Editor. The following individuals involved in the review of your submission have agreed to reveal their identity: Jason Andrews (Reviewer #1); Vegard Eldholm (Reviewer #2).

We agree that this paper is interesting, timely and raises important points. As is customary in *eLife*, the reviewers have discussed their critiques with one another. What follows below is the Reviewing Editor's edited compilation of the essential and ancillary points provided by reviewers in their critiques and in their interaction post-review. Please submit a revised version that addresses these concerns directly. Although we expect that you will address these comments in your response letter, we also need to see the corresponding revision clearly marked in the text of the manuscript. Some of the reviewers' comments may seem to be simple queries or challenges that do not prompt revisions to the text. Please keep in mind, however, that readers may have the same perspective as the reviewers. Therefore, it is essential that you attempt to amend or expand the text to clarify the narrative accordingly.

Essential revisions:

1) Both reviewers had some comments about the simulations for R0 or 1.1 and 0.9 – additional clarity on what impacts R0, and more substantively, can TBL and clustering rate help in estimating R0, if the infectious duration is held fixed? R1 notes "These issues should be addressed and/or the conclusions about lack of ability to infer R0 differences should be tempered a bit".

2) While both reviewers (and I) really like the idea of having an illustrative example showing how this could matter in a scenario with two co-circulating lineages in the same community, the differences seem too pronounced.

3) I have an additional comment (as reviewing editor):

You note that you select simulations with 100-2500 tips – how do you initialize the R0 < 1 simulations to end up with at least 100 tips? The tree size will presumably differ with different R0, as will the number of simulations you run that don't meet the size condition.

Given that you also sample in the last 10 years of the simulation. A very different fraction of the process will potentially end up in the last 10 years under the different parameters. Accordingly, the relationship between R0, TBL and clustering will be complex, and the fact that the results don't seem to be informative about R0 may be partly a result of this "simulation bias" and sampling period.

4) R1 notes that more comments on the sources of data that could help to disentangle the impact of differences in latency vs transmission would be helpful. Presumably that's a population-representative sampling that would enable knowing the fraction of cases that are of the two co-circulating types, over time? Or good estimates of the infectious duration? Or either, or both.

5) Please address minor points on clarity and communication raised by the reviewers. In addition, could you comment on the mutation rate during latency, and what might the result be if mutations were not occurring during latency at the rate that they do in active disease?

*Reviewer #1 (Recommendations for the authors):*

Overall, the manuscript was fairly clear and straightforward to interpret, with a few comments below. The main concern I had was with the overarching conclusion that TBLs are non-informative about R0. The results show that transmission rates are associated with TBL/clustering, but the argument is that R0 is not because it depends on the ratio between the transmission rate and removal rate, each of which can alter those metrics. But it would be reasonable to make assumptions about the infectious duration, for example within a lineage and/or location, enabling comparison between variants within a lineage, within lineage over time or space, etc. Holding infectious duration constant, comparisons of R0 would then be possible through TBL or clustering. Moreover, in some cases, the infectious duration can be directly estimated (i.e. by performing a prevalence study and comparing with notifications), enabling the relation of R0 to TBL/clustering directly through transmission rates. These issues should be addressed and/or the conclusions about lack of ability to infer R0 differences should be tempered a bit. This wouldn't diminish the importance of the study, as explaining the relationship between all of these parameters and these metrics is the valuable task that this article undertakes.

*Reviewer #2 (Recommendations for the authors):*

First, this is a cool and timely paper, and I really like the approach.

There are however some things I believe could be clarified and perhaps explored a little bit further.

Page 7 "An example": Drawing up an example of how different characteristics of co-circulating TB-types can result in contra-intuitive findings is great. In the example, a less transmissible type nevertheless has an R0 > 1 whereas a more transmissible variant has R0 < 1. Looking at table 2, it seems evident that the more transmissible Type 2 is contracting (R0<1) due to higher cure/death rates and higher sampling compared to the less transmissible Type 1, whereas it exhibits shorter TBLs and higher clustering due to shorter latency and a slower molecular clock compared to Type 1.

I have a few questions/comments which I hope will clear this up a little bit on my part:

- I think the example would be easier to follow if you explained which of the parameters influenced the R0, TBL and clustering rate, as I have tried to do above. If my summary is not correct, I guess that illustrates that this is a bit complex for many readers.

- This is a bit embarrassing, but what specifically should sampling rate be interpreted as here, and throughout the paper? If I get this right, sampling represents both observation and removal (the patient is sampled, and hence also cured and does not transmit further)?

- I think it would also be cool with a little description of the two types in more biological/medical lingo. If I understand correctly, Type 2 is more transmissible, and also characterized by rapid onset of disease (short latency), followed by rapid death, self-cure or health-seeking. This actually makes sense, and the only reason R0 < 1 is that health-seeking, cure and death rates are even more elevated than transmissibility?

Page 3: If I'm not wrong, the model used is a form of birth-death model? Perhaps this could be spelled out, and if not, explain how it differs from a B-D model.

Suppl figures: I believe the numbering of the suppl figures in relation to the text is wrong in quite a few instances

[Editors' note: further revisions were suggested prior to acceptance, as described below.]

Thank you for resubmitting your work entitled "Understanding drivers of phylogenetic clustering and terminal branch lengths distribution in epidemics of *Mycobacterium tuberculosis*" for further consideration by *eLife*. Your revised article has been evaluated by a Senior Editor and a Reviewing Editor.

The manuscript has been improved but there are some remaining issues that need to be addressed, as outlined below:

In response to the question about bias induced by having to choose among your simulations, I fear that you have misunderstood the question and therefore haven't addressed it. (The new exploration of the impact of sampling period is nice to have, though). You write: "Regarding the settings: all simulations are initialized as a single individual, simulations that result in less than 100 tips are discarded. For R0 < 1, a larger number of simulations needs to be discarded, as trees are tendentially smaller. "

This is a source of potentially large bias because the tree you obtain is likely not to meaningfully *have* an R0 of 0.9. To grow from 1 to 100, the mean number of offspring that is realised is definitely above 1 for an extended period, even if the parameter you set in the process was such that the *expected* number was less than 1. (I would be curious to see a birth-death simulation with an R0 of 0.9 that grows from 1 to 100 tips and last over 10 years). The probability of this growth is not zero, but it's small, so the trees you obtain are not at all representative of R0 = 0.9 trees. This means that when the clustering differs or fails to differ, you can't interpret that as being able to identify (or not) trees from R=0.9 vs R=1.1.

The tree sizes will be different too, and from coalescent theory, we know that this alone will change the patterns of genetic diversity (and hence the clustering). I would hazard that if you took simulations from identical parameters (birth, death, sampling, etc all the same) but in one set you chose the rejection criterion to have the trees need to grow bigger or last longer than in the other set, you would also find differences in clustering as a sole function of the cutoff criterion. Some portion of what you find is due probably due primarily to size and you could explore this, as well as not so dramatically biasing the trees you analyse by requiring extremely unlikely events before a tree gets into your sample.

Think of it the other way. If you took trees from a simulation with R0> 1 but you *only* looked at trees that went extinct before reaching 10 taxa, and then you did estimation or some other analysis on those trees, they would look very similar to trees with an R0 < 1 because those also die out. By removing the ones that grew you removed the information that R0 was set to be > 1. Conversely, here, by rejecting the vast majority of trees simulated under R0 < 1 that died out, you are removing the information that R0 was ever < 1.

I have asked for a revision because I think this point is not minor, as quite a bit of the paper is focused on clustering and R0 under simulations, and this same issue could impact many of them. It would be interesting to explore a genuinely declining population (rather than one that is declining in expectation because R0 < 1 but growing in its actual realisation because you start with 1 taxon and branch, rejecting those simulations that do not grow). However, initialising that population is challenging because the results would depend on the initial genetic diversity.

As a side note – under point 4 in the response, you note that "At least, in theory, all the parameters of the model can be estimated with phylodynamic analyses." But this paper finds a fundamental unidentifiability that suggests that these parameters are not identifiable: https://academic.oup.com/mbe/article/38/9/4010/6278301. They use likelihoods, not clustering, but since the likelihoods and portion clustered are fundamentally based on the branching times in the phylogenies it seems that their results would probably carry over.

---

## [Author Response]

Essential revisions:1) Both reviewers had some comments about the simulations for R0 or 1.1 and 0.9 – additional clarity on what impacts R0, and more substantively, can TBL and clustering rate help in estimating R0, if the infectious duration is held fixed? R1 notes "These issues should be addressed and/or the conclusions about lack of ability to infer R0 differences should be tempered a bit".

Regarding the main point (“can TBL and clustering rate help in estimating R0, if the infectious duration is held fixed?”):

In empirical studies, TBL and clustering rates can be used to estimate R0 only if the clock rate, the sampling proportion, the sampling period, the length of the infectious period, and the length of the latency period are ALL known or held fixed. If the infectious period is fixed, but the other factors are not, R0 would not necessarily correlate with clustering rates and TBL. To show this more clearly, I modified the example (Page 6), which now consists of two MTB “types” with identical sampling and death rates (and therefore identical infectious period), but different latency, transmission and clock rates. One type has R0 = 0.9, and the other has R0 = 1.1 (because of different transmission rates). Again, the shrinking type resulted in larger clustering rates and shorter TBL (because of differences in clock rates and latency). This is described in the response to point (2)

I also expanded the description of what influences R0 (Page 4, Ln 146-153):

R0 is the ratio between the rate at which infectious individuals are created (numerator) and the rate at which infectious individuals are removed (denominator). If the numerator is larger than the denominator the epidemic will grow (R0 > 1). Conversely, if the denominator is larger than the numerator the epidemic will shrink (R0 < 1). Specifically, with the epidemiological model used in this study the numerator is the transmission rate (λ), while the denominator is determined by the sum of the sampling rate and the death rate (*σ+ε*). R0 = λ / *(σ +ε).*

In non-mathematical terms R0 is determined by the relation between the transmission rate per unit of time and the average length of the infectious period (which is inversely proportional to the removal rate *σ +ε*). So that a large transmission rate (per unit of time) can result in a shrinking epidemic if the infectious period is very short. Conversely, a low transmission rate per unit of time can result in a growing epidemic if the infectious period is sufficiently long.

I think some confusion on these issues was generated by the order in which I presented the results. To make these points clearer I modified the structure of the Results section (Pages. 3-5). I also amended the abstract (Ln 21) and expanded the discussion (Page 7, Ln 234-242).

2) While both reviewers (and I) really like the idea of having an illustrative example showing how this could matter in a scenario with two co-circulating lineages in the same community, the differences seem too pronounced.

The main criticism of R1 was that in the example the length of the infectious period was drastically different between the two types, and that there is no direct evidence for differences of this magnitude between strains co-circulating in the same setting. I agree with R1 that the evidence for this is indirect. Therefore, I repeated the analysis assuming identical sampling and death rates for the two types (and therefore the same infectious period), also in this case the shrinking population resulted in higher clustering rates and shorter TBL, which were caused by shorter latency and lower clock rate (Page 6).

3) I have an additional comment (as reviewing editor):You note that you select simulations with 100-2500 tips – how do you initialize the R0 < 1 simulations to end up with at least 100 tips? The tree size will presumably differ with different R0, as will the number of simulations you run that don't meet the size condition.Given that you also sample in the last 10 years of the simulation. A very different fraction of the process will potentially end up in the last 10 years under the different parameters. Accordingly, the relationship between R0, TBL and clustering will be complex, and the fact that the results don't seem to be informative about R0 may be partly a result of this "simulation bias" and sampling period.

Thank you for this comment. In the first version of the manuscript, I did not consider a factor that could influence the results of the analysis: the sampling period.

To answer this comment, I performed an additional analysis. First, I tested whether different sampling periods can lead to differences in TBL and clustering rates (all else being equal). I tested three different values (5, 10, and 20 years of sampling), and found that indeed, shorter sampling periods resulted in lower clustering rates and longer terminal branches. This is probably because with shorter sampling periods the proportion of samples at the edges of the sampling window is larger. These samples are less likely to be clustered, as only one side of the transmission chain is present in the sample set (Page 3, Ln 108; Appendix 3).

The sampling period could potentially influence the analysis of scenarios with different R0. It is possible that scenarios with R0 < 1 result in generally shorter trees (because the lineage goes extinct within the first 10 years), potentially leading to a difference in the sampling period. Therefore, I repeated the analyses of transmission rates, infectious period, R0, and the example. In these new analyses I added an additional condition for the simulation to be accepted. If the tree height was lower than 10 years, the simulation was rejected. In this way I ensured that all simulations had similar sampling periods (~ 10 years). The results did not change, meaning that the sampling period did not bias the results of these analysis.

Regarding the settings: all simulations are initialized as a single individual, simulations that result in less than 100 tips are discarded. For R0 < 1, a larger number of simulations needs to be discarded, as trees are tendentially smaller.

4) R1 notes that more comments on the sources of data that could help to disentangle the impact of differences in latency vs transmission would be helpful. Presumably that's a population-representative sampling that would enable knowing the fraction of cases that are of the two co-circulating types, over time? Or good estimates of the infectious duration? Or either, or both.

At least in theory, all the parameters of the model can be estimated with phylodynamic analyses. However, this is quite challenging for MTB data sets. In addition to clustering rates and TBL, researchers can use complementary data to understand the epidemiological dynamics. For example, the change through time in the proportion of cases belonging to a certain type can be used as a proxy for the relative reproductive number (i.e., if a type is increasing in proportion within a population, it should have a larger R0 compared to other types). Similarly, if the absolute number of cases is increasing, R0 should be greater than one. However, there can be confounding factors, such as the migration of strains from other regions. Finally, the length of the infectious and latent period can be estimated (with some assumptions) from prevalence surveys and notification data. Usually this is done at the population level, the same analysis on individual lineages, or clones, could provide useful insights on the heterogeneity of MTB populations.

I included this paragraph in the discussion (Page 7, Ln 243-255).

5) Please address minor points on clarity and communication raised by the reviewers. In addition, could you comment on the mutation rate during latency, and what might the result be if mutations were not occurring during latency at the rate that they do in active disease?

I addressed all points raised by the reviewers (see below). Regarding latency and clock rates, first a clarification: here, with latency, I simply refer to the time between being infected and becoming infectious, not to a particular bacterial metabolic state (as often done in the TB literature). In any case, the effect of latency and clock rate are additive. For example: in a type with longer latency and lower clock rate during latency (which would lead to lower clock rate overall), the longer latency would result in lower clustering rates and longer TBL, while the lower clock rate has the opposite effect. The outcome will be determined by the magnitude of the changes: if the variation in clock rates is large enough to compensate for the longer latency, clustering rates will be larger and TBL shorter, otherwise clustering rates will be lower, and TBL longer.

I included this point in the discussion (Page 9, Ln 328-333).

Reviewer #1 (Recommendations for the authors):Overall, the manuscript was fairly clear and straightforward to interpret, with a few comments below. The main concern I had was with the overarching conclusion that TBLs are non-informative about R0. The results show that transmission rates are associated with TBL/clustering, but the argument is that R0 is not because it depends on the ratio between the transmission rate and removal rate, each of which can alter those metrics. But it would be reasonable to make assumptions about the infectious duration, for example within a lineage and/or location, enabling comparison between variants within a lineage, within lineage over time or space, etc. Holding infectious duration constant, comparisons of R0 would then be possible through TBL or clustering. Moreover, in some cases, the infectious duration can be directly estimated (i.e. by performing a prevalence study and comparing with notifications), enabling the relation of R0 to TBL/clustering directly through transmission rates. These issues should be addressed and/or the conclusions about lack of ability to infer R0 differences should be tempered a bit. This wouldn't diminish the importance of the study, as explaining the relationship between all of these parameters and these metrics is the valuable task that this article undertakes.

I addressed this in the answer to point (1) I report here again the main point.

In empirical studies, TBL and clustering rates could be used to estimate R0 only if the clock rate, the sampling proportion, the sampling period, the length of the infectious period, and the length of the latency period are ALL known or held fixed. If the infectious period is fixed, but the other factors are not, R0 would not necessarily correlate with clustering rates and TBL. To show this more clearly, I modified the example, which now consists of two types with identical infectious periods, but different latency, transmission, and clock rates. One type has R0 = 0.9, and the other has R0 = 1.1 (because of different transmission rates). The shrinking type resulted in higher clustering rates and shorter TBL (because of differences in clock rates and latency). This is described in the response to point (2)

I think some confusion on these issues was generated by the order in which I presented the results. To make these points clearer I modified the structure of the Results section (Pages. 3-5). I also amended the abstract (Ln 21) and expanded the discussion (Page 7, Ln 234-242).

Reviewer #2 (Recommendations for the authors):First, this is a cool and timely paper, and I really like the approach.There are however some things I believe could be clarified and perhaps explored a little bit further.Page 7 "An example": Drawing up an example of how different characteristics of co-circulating TB-types can result in contra-intuitive findings is great. In the example, a less transmissible type nevertheless has an R0 > 1 whereas a more transmissible variant has R0 < 1. Looking at table 2, it seems evident that the more transmissible Type 2 is contracting (R0<1) due to higher cure/death rates and higher sampling compared to the less transmissible Type 1, whereas it exhibits shorter TBLs and higher clustering due to shorter latency and a slower molecular clock compared to Type 1.I have a few questions/comments which I hope will clear this up a little bit on my part:- I think the example would be easier to follow if you explained which of the parameters influenced the R0, TBL and clustering rate, as I have tried to do above. If my summary is not correct, I guess that illustrates that this is a bit complex for many readers.

The summary of the original example is correct, there is only one missing aspect: the shorter TBL and larger clustering rates of Type 2 are indeed caused by shorter latency and lower clock rate. But, the larger transmission rate contributed to these trends as well.

In response to points 2 and 9 I changed the settings of the example, which is now simpler. In the updated version, the differences in R0 are due to different transmission rates, because the sampling and death rates are identical between the two types (therefore they have also identical infectious period). The shorter TBL and larger clustering rates of type 2 are now due to its shorter latency and lower clock rate (while the transmission rate is contrasting this trend).

I reworked the explanation of the example, because in the new version the two types are more similar to each other, it should be clearer what factors are influencing R0, TBL and clustering rates (Page 6).

- This is a bit embarrassing, but what specifically should sampling rate be interpreted as here, and throughout the paper? If I get this right, sampling represents both observation and removal (the patient is sampled, and hence also cured and does not transmit further)?

Yes, correct, the sampling rate *ε* is the rate at which infectious individuals are diagnosed and treated. It is assumed that these individuals stop being infectious immediately, hence they are removed from compartment I.

I added this information to the text (Page 3, Ln 92-94).

- I think it would also be cool with a little description of the two types in more biological/medical lingo. If I understand correctly, Type 2 is more transmissible, and also characterized by rapid onset of disease (short latency), followed by rapid death, self-cure or health-seeking. This actually makes sense, and the only reason R0 < 1 is that health-seeking, cure and death rates are even more elevated than transmissibility?

Yes, this was correct for the original analysis. However, it was pointed by R1 that there is no direct evidence for such divergent sampling and death rates within the same community. This is a fair criticism, I personally think that different durations of the infectious period (together with different transmission rates) could in part explain different TBL and clustering rates between e.g., L1 and L2, but the evidence supporting this is indeed indirect.

In any case, I clarified the differences between the two types in the text (Page 6).

Page 3: If I'm not wrong, the model used is a form of birth-death model? Perhaps this could be spelled out, and if not, explain how it differs from a B-D model.

Yes, it is a BD model, I added this information (Page 3, Ln 88).

Suppl figures: I believe the numbering of the suppl figures in relation to the text is wrong in quite a few instances

Fixed it. Thanks, and sorry for this.

[Editors' note: further revisions were suggested prior to acceptance, as described below.]

The manuscript has been improved but there are some remaining issues that need to be addressed, as outlined below:In response to the question about bias induced by having to choose among your simulations, I fear that you have misunderstood the question and therefore haven't addressed it. (The new exploration of the impact of sampling period is nice to have, though). You write: "Regarding the settings: all simulations are initialized as a single individual, simulations that result in less than 100 tips are discarded. For R0 < 1, a larger number of simulations needs to be discarded, as trees are tendentially smaller. "This is a source of potentially large bias because the tree you obtain is likely not to meaningfully *have* an R0 of 0.9. To grow from 1 to 100, the mean number of offspring that is realised is definitely above 1 for an extended period, even if the parameter you set in the process was such that the *expected* number was less than 1. (I would be curious to see a birth-death simulation with an R0 of 0.9 that grows from 1 to 100 tips and last over 10 years). The probability of this growth is not zero, but it's small, so the trees you obtain are not at all representative of R0 = 0.9 trees. This means that when the clustering differs or fails to differ, you can't interpret that as being able to identify (or not) trees from R=0.9 vs R=1.1.

Thank you for clarifying your comment, which indeed I had misunderstood in the first revision. This is an important point, before addressing it (see below), I want to clarify one detail regarding the simulations. To be accepted a simulation does not need to grow to 100 tips *existing at the same time*. Rather, 100 individuals must have existed *over the whole simulated period*. Therefore, the simulated lineages are not necessarily growing strongly, in fact some may go extinct before the end of the simulated period. These settings represent a much-reduced source of potential bias compared to what is written in the comment above.

I clarified the description of the simulation setup to avoid misunderstandings (page 10):

“A post-simulation condition was implemented by specifying the minimum and maximum number of tips in the phylogenetic tree necessary to accept the simulation. This threshold regards the total number of tips sampled during the simulated period, not the number of lineages existing at the most recent time point.”

The tree sizes will be different too, and from coalescent theory, we know that this alone will change the patterns of genetic diversity (and hence the clustering). I would hazard that if you took simulations from identical parameters (birth, death, sampling, etc all the same) but in one set you chose the rejection criterion to have the trees need to grow bigger or last longer than in the other set, you would also find differences in clustering as a sole function of the cutoff criterion.

I partially investigated this already in the first version of this manuscript (see Results at pages 3-4, and Appendix 4), although only for R0 = 1. In that analysis I found that different cutoff values on the minimum tree size did not change clustering rates or TBL. I now expanded this analysis and performed it also for different values of R0 confirming that the results are robust to different thresholds on the minimum tree size (see answer to next point).

Some portion of what you find is due probably due primarily to size and you could explore this, as well as not so dramatically biasing the trees you analyse by requiring extremely unlikely events before a tree gets into your sample.Think of it the other way. If you took trees from a simulation with R0> 1 but you *only* looked at trees that went extinct before reaching 10 taxa, and then you did estimation or some other analysis on those trees, they would look very similar to trees with an R0 < 1 because those also die out. By removing the ones that grew you removed the information that R0 was set to be > 1. Conversely, here, by rejecting the vast majority of trees simulated under R0 < 1 that died out, you are removing the information that R0 was ever < 1.

I expanded the analyses investigating transmission and sampling rates (Appendix 5). I focused on these because they are the ones exploring the broadest range of R0 values (from 0.8 to 1.2). With the original settings I accepted only simulations that resulted in at least 100 tips over the whole simulated period. I repeated these analyses using 50 and 25 as alternative minimum tips thresholds, with these settings the potential bias should be reduced. In addition, I included a scenario with a larger threshold (200 tips), to observe how the results changed when increasing the potential source of bias.

For both transmission and sampling rates I confirmed the results of the original analysis for all thresholds. Increasing transmission rates (and therefore R0) resulted in larger clustering rates and shorter terminal branches. This trend was clear for all thresholds, and it was monotone over the entire range of R0 values (0.8-1.2). Conversely, the sampling rate did not strongly affect clustering rates and TBL. There might be a weak trend towards lower clustering rates for lower sampling rates, this was similar for all thresholds, and it was mostly visible for R0 values between 0.8 and 1 (Appendix 5 – figures 1-8, Appendix 5 – tables 1-8).

To explore the influence of different thresholds more directly, I compared the results obtained with the same settings and different thresholds. To limit the number of supplementary tables and figures I focused on the scenarios with R0 = 0.8, 1, and 1.2. When I compared the results obtained for R0 = 0.8, the scenario with potentially the largest bias, I found that the clustering rates and TBL were not identical when using different thresholds on the minimum tree size. There was a weak trend towards lower clustering rates for larger thresholds (on the minimum tree size), although this was not monotone for all SNP thresholds (Appendix 5 – figures 9-10, Appendix 5 – table 9-10). Additionally, for one of the analyses with R0 = 0.8 (λ = 1 and *ε =* 1.25) I found that larger thresholds also resulted in longer terminal branches, although the difference between the mean of the TBL distributions was smaller than 0.01 SNPs (0.3729, 0.3768, 0.3773, and 0.3815 for a threshold of 25, 50, 100, and 200 respectively). These trends were not present for larger values of R0, indicating that they are indeed a bias due to the minimum tree size, which affect scenarios with shrinking populations (Appendix 5 – figures 11-13, Appendix 5 – tables 11-13).

These results suggest that the threshold on the minimum tree size can bias the outcome of some analyses. However, with the settings used in this study this bias was negligible, and the results were robust to different thresholds. For the transmission rate, the trend towards larger clustering rates for higher transmission rates was clear also for scenarios with R0 equal or greater than one, these scenarios are scarcely affected by the bias, further indicating that this is a genuine trend due to the epidemiological process (Appendix 5 – figures 1 and 3-5). Conversely, for the sampling rate the observed trend was evident mostly for scenarios with R0 smaller than one, suggesting that it was caused by the threshold on the tree size (Appendix 5 – figures 2 and 6-8). In any case the differences were rather small, and I conclude that if there is any effect of the sampling rate on clustering and TBL, this is extremely weak.

I report all these new analyses in Appendix 5 (page 21). I also modified the manuscript (pages 4-5) to include the latest results.

I have asked for a revision because I think this point is not minor, as quite a bit of the paper is focused on clustering and R0 under simulations, and this same issue could impact many of them.

I agree that these aspects needed to be explored before publication. I would also like to point out that there are several factors that are not related to transmission, or R0, but influence the results of clustering and TBL (the most relevant are latency and the molecular clock rate). Therefore, this study would reach the same conclusion (i.e., that clustering rates and TBL cannot be used to identify differences in transmission or evolutionary success, unless all other factors are known or identical between sub-populations), also without including the analyses of scenarios with R0 < 1. Of course, it is much better to include these analyses, as they allow to explore the effect of the transmission and sampling rates, and therefore R0, on clustering rates and TBL, and I’m glad I could show that they are robust to different thresholds on the minimum tree size.

It would be interesting to explore a genuinely declining population (rather than one that is declining in expectation because R0 < 1 but growing in its actual realisation because you start with 1 taxon and branch, rejecting those simulations that do not grow). However, initialising that population is challenging because the results would depend on the initial genetic diversity.

The editor already identified a fundamental challenge. In a stable population (R0 = 1), 50% of the lineages existing at any time point are expected to die without transmitting further, with R0 = 0.8 this proportion rises to 60%. Therefore, if we initialize a population and consider all individuals, the clustering rate will be in large part determined by the initial genetic diversity, which has nothing to do with the simulated epidemiological processes. It would be possible to repeat this analysis several times with different “starting populations”, characterized by different underlying genealogies. However, the set of possible starting conditions is extremely large, and I believe that these analyses could not alter the main findings of this study.

As a side note – under point 4 in the response, you note that "At least, in theory, all the parameters of the model can be estimated with phylodynamic analyses." But this paper finds a fundamental unidentifiability that suggests that these parameters are not identifiable: https://academic.oup.com/mbe/article/38/9/4010/6278301. They use likelihoods, not clustering, but since the likelihoods and portion clustered are fundamentally based on the branching times in the phylogenies it seems that their results would probably carry over.

Thank you for pointing this out. I removed that sentence from the manuscript, and I amended the conclusions (pages 9-10), to take into account these findings:

“Phylodynamic methods that estimate the parameters of an epidemiological model from genomic data are becoming available (Kühnert et al. 2016, Didelot et al. 2017, Volz and Siveroni 2018), however, these analyses have limits that cannot be overcome exclusively with phylogenetic data (Louca et al. 2021), and in any case they are challenging with current MTB data sets (Kühnert et al. 2018, Walter et al. 2022). Methodological advances, the integration of different types of data, and more complete and longer sampling series of MTB epidemics will allow to study epidemiological dynamics more accurately in the future. In the meantime, the results of clustering and TBL analyses should not be over-interpreted.”